# Approximate Heavy Tails in Offline (Multi-Pass) Stochastic Gradient Descent

**Krunoslav Lehman Pavasovic**
Inria Paris, CNRS, Ecole Normale Supérieure, PSL Research University
Paris, France
krunoslav.lehman-pavasovic@inria.fr

**Alain Durmus**
CMAP, CNRS, Ecole Polytechnique, Institut Polytechnique de Paris
Paris, France
alain.durmus@polytechnique.edu

**Umut Şimşekli**
Inria Paris, CNRS, Ecole Normale Supérieure, PSL Research University
Paris, France
umut.simsekli@inria.fr

## Abstract

A recent line of empirical studies has demonstrated that SGD might exhibit a heavy-tailed behavior in practical settings, and the heaviness of the tails might correlate with the overall performance. In this paper, we investigate the emergence of such heavy tails. Previous works on this problem only considered, up to our knowledge, *online* (also called single-pass) SGD, in which the emergence of heavy tails in theoretical findings is contingent upon access to an infinite amount of data. Hence, the underlying mechanism generating the reported heavy-tailed behavior in practical settings, where the amount of training data is finite, is still not well-understood. Our contribution aims to fill this gap. In particular, we show that the stationary distribution of *offline* (also called multi-pass) SGD exhibits 'approximate' power-law tails and the approximation error is controlled by how fast the empirical distribution of the training data converges to the true underlying data distribution in the Wasserstein metric. Our main takeaway is that, as the number of data points increases, offline SGD will behave increasingly 'power-law-like'. To achieve this result, we first prove nonasymptotic Wasserstein convergence bounds for offline SGD to online SGD as the number of data points increases, which can be interesting on their own. Finally, we illustrate our theory on various experiments conducted on synthetic data and neural networks.

## 1 Introduction

Many machine learning problems can be cast as the following population risk minimization problem:

$$\text{minimize } x \mapsto F(x) := \mathbb{E}[f(x, Z)] , \quad Z \sim \mu_z , \tag{1}$$

where $x \in \mathbb{R}^d$ denotes the model parameters, $\mu_z$ is the data distribution over the measurable space $(\mathsf{Z}, \mathcal{Z})$ and $f : \mathbb{R}^d \times \mathsf{Z} \to \mathbb{R}$ is a loss function. The main difficulty in addressing this problem is that $\mu_z$ is typically unknown. Suppose we have access to an *infinite* sequence of independent and

37th Conference on Neural Information Processing Systems (NeurIPS 2023).

identically distributed (i.i.d.) data samples $\mathcal{D} := \{Z_1, Z_2, \dots\}$ from $\mu_z$. In that case, we can resort to the *online* stochastic gradient descent (SGD) algorithm, which is based on the following recursion:

$$X_{k+1} = X_k - \frac{\eta}{b} \sum_{i \in \Omega_{k+1}} \nabla f(X_k, Z_i) , \tag{2}$$

where $k$ denotes the iterations, $\Omega_k := \{b(k-1)+1, b(k-1)+2, \dots, bk\}$ is the batch of data-points at iteration $k$, $\eta$ denotes the step size, $b$ denotes the batch size such that $|\Omega_k| = b$, and $Z_i$ with $i \in \Omega_k$, denotes the $i$-th data sample at the $k$-th iteration. This algorithm is also called 'single-pass' SGD, as it sees each data point $Z_i$ in the infinite data sequence $\mathcal{D}$ only once.

While drawing i.i.d. data samples at each iteration is possible in certain applications, in the majority of practical settings, we only have access to a finite number of data points, preventing online SGD use. More precisely, we have access to a dataset of $n$ i.i.d. points $\mathcal{D}_n := \{Z_1, \dots, Z_n\}$[1], and given these points the goal is then to minimize the empirical risk $\hat{F}^{(n)}$, given as follows:

$$\text{minimize } x \mapsto \hat{F}^{(n)}(x) := \frac{1}{n} \sum_{i=1}^n f(x, Z_i) \text{ over } \mathbb{R}^d . \tag{3}$$

To attack this problem, one of the most popular approaches is the *offline* version of (2), which is based on the following recursion:

$$X_{k+1}^{(n)} = X_k^{(n)} - \frac{\eta}{b} \sum_{i \in \Omega_{k+1}^{(n)}} \nabla f(X_k^{(n)}, Z_i) , \tag{4}$$

where $\Omega_k^{(n)} \subset \{1, \dots, n\}$ denotes the indices of the (uniformly) randomly chosen data points at iteration $k$ with $|\Omega_k^{(n)}| = b \leq n$ and $\cdot^{(n)}$ emphasizes the dependence on the sample size $n$. Analogous to the single-pass regime, this approach is also called the *multi-pass* SGD, as it requires observing the same data points multiple times.

Despite its ubiquitous use in modern machine learning applications, the theoretical properties of offline SGD have not yet been well-established. Among a plethora of analyses addressing this question, one promising approach has been based on the observation that parameters learned by SGD can exhibit *heavy tails*. In particular, [ŞGN+19] have empirically demonstrated a heavy-tailed behavior for the sequence of stochastic gradient noise:

$$\left( \nabla \hat{F}^{(n)}(X_k^{(n)}) - b^{-1} \sum_{i \in \Omega_{k+1}^{(n)}} \nabla f(X_k^{(n)}, Z_i) \right)_{k \geq 1} .$$

Since then, further studies have extended this observation to other sequences appearing in machine learning algorithms [ZFM+20, MM19, ZLMU22]. These results were recently extended by [BSE+21], who showed that the parameter sequence $(X_k^{(n)})_{k \geq 1}$ itself can also exhibit heavy tails for large $\eta$ and small $b$. These empirical investigations all hinted a connection between the observed heavy tails and the generalization performance of SGD: heavier the tails might indicate a better generalization performance.

Motivated by these empirical findings, several subsequent papers theoretically investigated how heavy-tailed behavior can emerge in stochastic optimization. In this context, [GSZ21] have shown that when *online SGD* is used with a quadratic loss (i.e., $f(x, z = (a, y)) = 2^{-1}(a^\top x - y)^2$), the distribution of the iterates $(X_k)_{k \geq 0}$ can converge to a heavy-tailed distribution, even with exponentially light-tailed data (e.g., Gaussian). More precisely, for a fixed step-size $\eta > 0$, they showed that, under appropriate assumptions, there exist constants $c \in \mathbb{R}_+$ and $\alpha > 0$, such that the following identity holds:

$$\lim_{t \to \infty} t^\alpha \mathbb{P}\left( \|X_\infty\| > t \right) = c, \tag{5}$$

where $X_\infty \sim \pi$ and $\pi$ denotes the stationary distribution of online SGD (2). This result illustrates that the distribution of the online SGD iterates follows a power-law decay with the *tail index* $\alpha$: $\mathbb{P}\left( \|X_\infty\| > t \right) \approx t^{-\alpha}$ for large $t$. Furthermore, $\alpha$ depends monotonically on the algorithm

---

[1]Here we deliberately choose the same notation $Z_i$ for both infinite and finite data regimes to highlight the fact that we can theoretically view the finite data regime from the following perspective: given an infinite sequence of data points $\mathcal{D}$, the finite regime only uses *the first $n$ elements* of $\mathcal{D}$, which constitute $\mathcal{D}_n$.

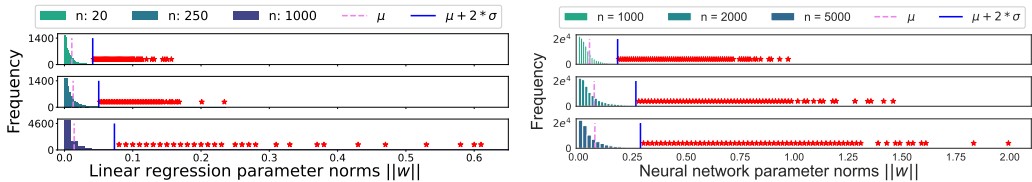

Figure 1: Left: histograms of the parameter norms for Gaussian data linear regression. Right: histograms of NN parameter norms for the first layer, trained on MNIST. Norms exceeding the $\mu + 2\sigma$ threshold are marked with a red asterisk.

hyperparameters $\eta$, $b$, and certain properties of the data distribution $\mu_z$. This result is also consistent with [DMN+21, Example 1], which shows a simple instance from linear stochastic approximation such that for any fixed step-size $\eta$, there exists $p_c > 0$ such that for any $p \geq p_c$, $\lim_{k \to +\infty} \mathbb{E}[\|X_k\|^p] = +\infty$. In a concurrent study, [HM21] showed that this property is, in fact, not specific for quadratic problems and can hold for more general loss functions and different choices of stochastic optimizers, with the constraint that i.i.d. data samples are available at every iteration[2].

Another line of research has investigated the theoretical links between heavy tails and the generalization performance of SGD. Under different theoretical settings and based on different assumptions, [SSDE20, BSE+21, LWŞ22, RBG+22, HSKM22, RZGŞ23] proved generalization bounds (i.e., bounds on $|\hat{F}^{(n)}(x) - F(x)|$) illustrating that the heavy tails can be indeed beneficial for better performance. However, all these results rely on *exact* heavy tails, which, to our current knowledge, can only occur in the *online* SGD regime where there is access to an infinite sequence of data points. Hence the current theory (both on the emergence of heavy tails and their links to generalization performance) still falls short in terms of explaining the empirical heavy-tailed behavior observed in *offline* SGD as reported in [ŞGN+19, MM19, ZFM+20, BSE+21].

**Main problematic and contributions.** Although empirically observed, it is currently unknown how heavy-tailed behavior arises in offline SGD since the aforementioned theory [GSZ21, HM21] requires infinite data. Our main goal in this paper is hence to develop a theoretical framework to catch *how* and *in what form* heavy tails may arise in offline SGD, where the dataset is finite.

We build our theory based on two main observations. The first observation is that, since we have finitely many data points in offline SGD, the moments (of any order) of the iterates $X_k^{(n)}$ can be bounded (see, e.g., [CGZ19]), hence in this case we cannot expect *exact* power-law tails in the form of (5) in general. Our second observation, on the other hand, is that since the finite dataset $\mathcal{D}_n$ can be seen as the first $n$ elements of the infinite sequence $\mathcal{D}$, as $n$ increases the offline SGD recursion (4) should converge to the online SGD recursion (2) in some sense. Hence, when online SGD shows a heavy-tailed behavior (i.e., the case where $n \to +\infty$), as we increase $n$, we can expect that the tail behavior of offline SGD should be more and more 'power-law-like'.

To further illustrate this observation, as a preliminary exploration, we run offline SGD in a 100-dimensional linear regression problem, as well as a classification problem on the MNIST dataset, using a fully-connected, 3-layer neural network.[3]

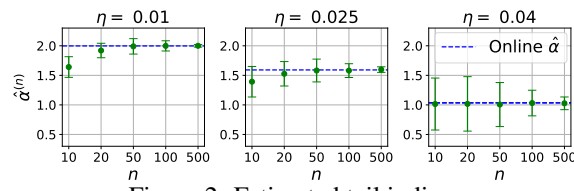

Figure 2: Estimated tail indices

In Figure 1, we plot the histograms of the parameter norms after 1000 (and 5000 resp.) iterations. We observe the following consistent patterns: *(i)* the means and standard deviations of the parameters' estimated stabilizing distributions increase with $n$, and *(ii)*, the quantity and magnitude of the parameters far from the bulk of the distribution increase with $n$. In other words, as we increase the number of samples $n$ in offline SGD, the behavior of the iterates becomes heavier-tailed.

---

[2]We will present these results in more detail in Section 2.

[3]The exact experimental details can be found in Section 4.

As another example, we utilize the tail-estimator from [GSZ21][4] to compare the estimated tail indices $\alpha$ for offline and online SGD. The results are plotted in Figure 2 and correspond to 10 random initializations with different step-sizes (for further details, see Tail estimation). We observe that, as $n$ increases, the estimated tail indices for offline SGD (shown in green) get closer to the true tail index corresponding to online SGD (the horizontal blue line).

Our main contribution in this paper is to make these observations rigorous and explicit. In particular, we extend the tail estimation results for online SGD [GSZ21, HM21] to offline SGD, and show that the stationary distribution of offline SGD iterates will exhibit *'approximate'* heavy tails. Informally, for both quadratic and a class of strongly convex losses, we show that, with high probability, there exist constants $c_1, c_2$, such that for large enough $t$ we have the following tail bound:

$$\left[ \frac{c_1}{t^\alpha} - \frac{1}{t} \mathbf{W}_1(\mu_z, \mu_z^{(n)}) \right] \lesssim \mathbb{P}(\|X_\infty^{(n)}\| > t) \lesssim \left[ \frac{c_2}{t^\alpha} + \frac{1}{t} \mathbf{W}_1(\mu_z, \mu_z^{(n)}) \right], \tag{6}$$

where $\alpha$ is the tail index of *online* SGD as given in (5), $X_\infty^{(n)} \sim \pi^{(n)}$ denotes a sample from the stationary distribution $\pi^{(n)}$ of *offline* SGD (4), $\mu_z^{(n)} = n^{-1} \sum_{i=1}^n \delta_{Z_i}$ denotes the empirical measure of the data points in the data sample $\mathcal{D}_n = \{Z_1, \ldots, Z_n\}$, and $\mathbf{W}_1$ denotes the Wasserstein-1 distance (to be defined formally in the next section).

Our result indicates that the tail behavior of offline SGD is mainly driven by two terms: (i) a persistent term $(c/t^\alpha)$ that determines the power-law decay, and (ii) a vanishing term $\mathbf{W}_1(\mu_z, \mu_z^{(n)})/t$, as we take $n \to \infty$. In other words, compared to (5), we can see that, with high-probability, $X_\infty^{(n)}$ exhibits an *approximate* power-law decay behavior, with a discrepancy controlled by $\mathbf{W}_1(\mu_z, \mu_z^{(n)})$.

Fortunately, the term $\mathbf{W}_1(\mu_z, \mu_z^{(n)})$ has been well-studied in the literature. By combining our results with Wasserstein convergence theorems for empirical measures [FG15], we further present nonasymptotic tail bounds for offline SGD. Our main takeaway is as follows: as $n$ increases, offline SGD will exhibit approximate power-laws, where the approximation error vanishes with a rate given by how fast the empirical distribution of the data points $\mu_z^{(n)}$ converges to the true data distribution $\mu_z$ as $n \to \infty$.

To prove our tail bounds, as an intermediate step, we prove nonasymptotic Wasserstein convergence results for offline SGD in the form: $\mathbf{W}_1(\pi, \pi^{(n)}) \lesssim \mathbf{W}_1(\mu_z, \mu_z^{(n)})$, with high probability, recalling that $\pi$ and $\pi^{(n)}$ respectively denote online and offline SGD stationary distributions. These results, we believe, hold independent interest for the community. Finally, we support our theory with various experiments conducted on both synthetic quadratic optimization problems and real neural network settings.

## 2 Preliminaries and Technical Background

### 2.1 Notation and preliminary definitions

We let $\mathrm{I}_d$ denote the $d \times d$ identity matrix. We denote $[n] = \{1, \ldots, n\}$, $\mathscr{P}_{n,b} = \{\mathsf{I} \subset [n], |\mathsf{I}| = b\}$. For a vector $x \in \mathbb{R}^d$, $\|x\|$ denotes the Euclidean norm of $x$, and for matrix $\mathbf{A}$, the norm $\|\mathbf{A}\| = \sup_{\|x\|=1} \|\mathbf{A}x\|$ denote its spectral norm. We denote with $\sigma_{\min}(\mathbf{A}), \sigma_{\max}(\mathbf{A})$ the smallest and largest singular values of $\mathbf{A}$, respectively. $\mathcal{P}(\mathbb{R}^d)$ is the set of all probability measures on $\mathbb{R}^d$, and $\mathscr{L}(X)$ denotes the probability law of a random variable $X$. We denote the set of all couplings of two measures $\mu$ and $\nu$ with $\Gamma(\mu, \nu)$. Finally, for two functions $f(n)$ and $g(n)$ defined on $\mathbb{R}_+$, we denote $f = \mathcal{O}(g)$, if there exist constants $c \in \mathbb{R}_+$, $n_0 \in \mathbb{N}$, such that $f(n) \leq cg(n), \forall n > n_0$. The Dirac measure concentrated at $x$ is defined as follows: for a measurable set $E$, $\delta_x(E) = 1$ if $x \in E$, $\delta_x(E) = 0$, otherwise.

For $p \in [1, \infty)$, the Wasserstein-$p$ distance between two distributions $\mu$ and $\nu$ on $\mathbb{R}^d$ is defined as:

$$\mathbf{W}_p(\mu, \nu) := \inf_{\gamma \in \Gamma(\mu, \nu)} \left( \int \|x - y\|^p \, \mathrm{d}\gamma(x, y) \right)^{1/p}.$$

We correspondingly define the Wasserstein-$p$ distance between two distributions $\mu$ and $\nu$ on $\mathbb{R}^{d \times d}$ as

$$\mathbf{W}_p(\mu, \nu) := \inf_{\gamma \in \Gamma(\mu, \nu)} \left( \int \|\mathbf{A} - \mathbf{B}\|^p \, \mathrm{d}\gamma(\mathbf{A}, \mathbf{B}) \right)^{1/p}.$$

---

[4]The justification for the estimator choice can be found in the appendix, Sec. E.

## 2.2 Heavy tails in online SGD

We first formally define heavy-tailed distributions with a power-law decay, used throughout our work.

**Definition 1** (Heavy-tailed distributions with a power-law decay). *A random vector $X$ is said to follow a distribution with a power-law decay if $\lim_{t \to \infty} t^\alpha \mathbb{P}(\|X\| \geq t) = c_0$, for constants $c_0 > 0$ and $\alpha > 0$, the latter known as the tail index.*

The tail index $\alpha$ determines the thickness of the tails: larger values of $\alpha$ imply a lighter tail.

Recently, [GSZ21] showed that online SGD might produce heavy tails even with exponentially light-tailed data. More precisely, they focused on a quadratic optimization problem, where $z \equiv (a, q) \in \mathbb{R}^{d+1}$ and $f(x, z) = 2^{-1}(a^\top x - q)^2$. With this choice, the online SGD recursion reads as follows:

$$X_{k+1} = X_k - \frac{\eta}{b} \sum_{i \in \Omega_{k+1}} \left( a_i a_i^\top X_k - a_i q_i \right) = (\mathrm{I}_d - \eta \mathbf{A}_{k+1}) X_k + \eta \mathbf{b}_{k+1} \tag{7}$$

where $(Z_k)_{k \geq 1} \equiv (a_k, q_k)_{k \geq 1}$, $\mathbf{A}_k := b^{-1} \sum_{i \in \Omega_k} a_i a_i^\top$, and $\mathbf{b}_k := b^{-1} \sum_{i \in \Omega_k} a_i q_i$.

In this setting, they proved the following theorem.

**Theorem 1** ([GSZ21]). *Assume that $(a_k, q_k)_{k \geq 1}$ are i.i.d. random variables such that $a_1 \sim \mathrm{N}(0, \sigma^2 \mathrm{I}_d)$, with $\sigma^2 > 0$ and $q_i$ has a continuous density with respect to the Lebesgue measure on $\mathbb{R}$ with all its moments being finite. In addition suppose that $\mathbb{E}[\log \|\mathrm{I}_d - \eta \mathbf{A}_1\|] < 0$. Then, there exists a unique $\alpha > 0$ such that $\mathbb{E}[\|\mathrm{I}_d - \eta \mathbf{A}_1\|^\alpha] = 1$ and the iterates $(X_k)_{k \geq 0}$ in (7) converge in distribution to a unique stationary distribution $\pi$ such that if $X_\infty \sim \pi$ and $\alpha \notin \mathbb{N}$:*

$$\lim_{t \to \infty} t^\alpha \mathbb{P}(\|X_\infty\| > t) = c, \quad \text{with } c \in \mathbb{R}_+ . \tag{8}$$

The authors of [GSZ21] also showed that the tail index $\alpha$ is monotonic with respect to the algorithm hyperparameters $\eta$ and $b$: a larger $\eta$ or smaller $b$ indicates heavier tails, i.e., smaller $\alpha$[5].

In a concurrent study, [HM21] considered a more general class of loss functions and choices of stochastic optimization algorithms. They proved a general result, where we translate it for online SGD applied on strongly convex losses in the following theorem.

**Theorem 2** ([HM21]). *Assume that for every $z \in \mathsf{Z}$, $f(\cdot, z)$ is twice differentiable and is strongly convex. Consider the recursion in (2) with a sequence of i.i.d. random variables $(Z_k)_{k \geq 1}$ and let $R$ and $r$ be two non-negative functions defined as: for any $z \in \mathsf{Z}$,*

$$R(z) := \sup_{x \in \mathbb{R}^d} \left\| \mathrm{I}_d - \eta \nabla^2 f(x, z) \right\|, \text{ and } r(z) := \liminf_{\|x\| \to \infty} \sigma_{\min} \left( \mathrm{I}_d - \eta \nabla^2 f(x, z) \right) .$$

*Further assume that $\mathbb{E}[R(Z_1) + \|\nabla f(x^\star, Z_1)\|] < +\infty$, for some $x^\star \in \mathbb{R}^d$, $\mathbb{E}[\log R(Z_1)] < 0$, and $\mathbb{P}(r(Z_1) > 1) > 0$. Then, the iterates $(X_k)_{k \geq 1}$ in (2) admit a stationary distribution $\pi$. Moreover, there exist $\alpha, \beta > 0$ such that $\mathbb{E}[r(Z)^\alpha] = 1$, $\mathbb{E}[R(Z)^\beta] = 1$, and for any $\varepsilon > 0$:*

$$\limsup_{t \to \infty} t^{\alpha + \varepsilon} \mathbb{P}(\|X_\infty\| > t) > 0, \quad \text{and} \quad \limsup_{t \to \infty} t^{\beta - \varepsilon} \mathbb{P}(\|X_\infty\| > t) < +\infty ,$$

*where $X_\infty \sim \pi$.*

The result illustrates that power laws can arise in general convex stochastic optimization algorithms. We note that in general, Theorem 2 does not require *each* $f(\cdot, z)$ to be strongly convex, in fact, it can accommodate non-convex functions as well. However, we are not aware of any popular non-convex machine learning problem that can satisfy all the assumptions of Theorem 2. Nevertheless, for better illustration, in Section F, we show that an $\ell_2$-regularized logistic regression problem with random regularization coefficients falls into the scope of Theorem 2.

The two aforementioned theorems are limited to the online setting, and it is unclear whether they apply to a finite number of data points $n$. In the subsequent section, we aim to extend these theorems to the offline setting.

---

[5]Without the Gaussian data assumption, [GSZ21] proved lower-bounds over $\alpha$ instead of the exact identification of $\alpha$ as given in (8). We also note that the condition $\alpha \notin \mathbb{N}$ is not required in [GSZ21]: without such a condition, one can show that, for any $u \in \mathbb{R}^d$ with $\|u\| = 1$, $\lim_{t \to \infty} t^\alpha \mathbb{P}(u^\top X_\infty > t)$ has a non-trivial limit. For the clarity of the presentation, we focus on the tails of the norm $\|X_\infty\|$, hence use a non-integer $\alpha$, which leads to (8) (for the equivalence of these expressions, see [BDM+16, Theorem C.2.1]).

## 3  Main results

### 3.1  Quadratic objectives

We first focus on the quadratic optimization setting considered in Theorem 1 as its proof is simpler and more instructive. Similarly as before, let $(a_i, q_i)_{i \geq 1}$ be i.i.d. random variables in $\mathbb{R}^{d+1}$, such that $z_i \equiv (a_i, q_i)$. Following the notation from (4), we can correspondingly define the *offline* SGD recursion as follows:

$$X_{k+1}^{(n)} = X_k^{(n)} - \frac{\eta}{b} \sum_{i \in \Omega_{k+1}^{(n)}} \left( a_i a_i^\top X_k^{(n)} - a_i q_i \right) = (\mathrm{I}_d - \eta \mathbf{A}_{k+1}^{(n)}) X_k^{(n)} + \eta \mathbf{b}_{k+1}^{(n)}, \qquad (9)$$

where $\mathbf{A}_k^{(n)} := b^{-1} \sum_{i \in \Omega_k^{(n)}} a_i a_i^\top$, $\mathbf{b}_k^{(n)} := b^{-1} \sum_{i \in \Omega_k^{(n)}} a_i q_i$, and $\Omega_k^{(n)}$ is as defined in (4). Now, $(\mathbf{A}_k^{(n)}, \mathbf{b}_k^{(n)})_{k \geq 1}$ are i.i.d. random variables with respect to the empirical measure:

$$\mu_{\mathbf{A}, \mathbf{b}}^{(n)} = \binom{n}{b}^{-1} \sum_{i=1}^{\binom{n}{b}} \delta_{\{\overline{\mathbf{A}}_i^{(n)}, \overline{\mathbf{b}}_i^{(n)}\}}, \qquad (10)$$

where we enumerate all possible choices of minibatch indices $\mathscr{P}_{n,b} = \{\mathsf{I} \subset \{1, \ldots, n\} : |\mathsf{I}| = b\}$ as $\mathscr{P}_{n,b} = \{\mathsf{S}_1^{(n)}, \mathsf{S}_2^{(n)}, \ldots, \mathsf{S}_{\binom{n}{b}}^{(n)}\}$ and $(\overline{\mathbf{A}}_i^{(n)}, \overline{\mathbf{b}}_i^{(n)})_{i=1}^{\binom{n}{b}}$ are i.i.d. random variables defined as follows:

$$\overline{\mathbf{A}}_i^{(n)} = \frac{1}{b} \sum_{j \in \mathsf{S}_i^{(n)}} a_j a_j^\top, \quad \overline{\mathbf{b}}_i^{(n)} = \frac{1}{b} \sum_{j \in \mathsf{S}_i^{(n)}} a_j q_j. \qquad (11)$$

We denote by $\mu_{\mathbf{A}, \mathbf{b}}$ the common distribution of $(\overline{\mathbf{A}}_i^{(n)}, \overline{\mathbf{b}}_i^{(n)})_{i=1}^{\binom{n}{b}}$. Note that it does not depend on $n$ but only $b$.

With these definitions at hand, we define the two marginal measures of $\mu_{\mathbf{A}, \mathbf{b}}$ and $\mu_{\mathbf{A}, \mathbf{b}}^{(n)}$ with $\mu_{\mathbf{A}}$, $\mu_{\mathbf{b}}$, and $\mu_{\mathbf{A}}^{(n)}$, $\mu_{\mathbf{b}}^{(n)}$, respectively. Before proceeding to the theorem, we have to define Wasserstein ergodicity. A discrete-time Markov chain $(Y_k)_{k \geq 0}$ is said to be (Wasserstein-1) geometrically ergodic if it admits a stationary distribution $\pi_Y$ and there exist constants $c \geq 0$ and $\rho \in (0, 1)$, such that

$$\mathbf{W}_1(\mathscr{L}(Y_k), \pi_Y) \leq c e^{-\rho k}, \text{ for any } k \geq 0. \qquad (12)$$

The chain is simply called (Wasserstein-1) ergodic if $\lim_{k \to \infty} \mathbf{W}_1(\mathscr{L}(Y_k), \pi_Y) = 0$.

We can now state our first main contribution, an extension of Theorem 1 to the offline setting:

**Theorem 3.** *Let $n \geq 1$ and $\epsilon_n \in [0, 1]$ be the probability that $(X_k^{(n)})_{k \geq 0}$ is not ergodic (in the Wasserstein sense)[6] with a stationary distribution $\pi^{(n)}$ having finite q-th moment with $q > 1$. Assume that the conditions of Theorem 1 hold with $\alpha > 1$. Then, for any $\epsilon > 0$, there exist constants $\tilde{c}_1, \tilde{c}_2$, and $t_0 > 0$ such that for all $\zeta \in (0, 1]$, $t > t_0$, with probability larger than $1 - \epsilon_n - \zeta$, the following inequalities hold:*

$$\left[ \frac{1}{2^\alpha} \frac{c - \epsilon}{t^\alpha} - \frac{\tilde{c}_1}{t n^{1/2}} \sqrt{\log \frac{\tilde{c}_2}{\zeta}} \right] \leq \mathbb{P}(\|X_\infty^{(n)}\| > t) \leq \left[ 2^\alpha \frac{c + \epsilon}{t^\alpha} + \frac{2\tilde{c}_1}{t n^{1/2}} \sqrt{\log \frac{\tilde{c}_2}{\zeta}} \right], \qquad (13)$$

*where c is given in (8).*

This result shows that whenever the online SGD recursion admits heavy tails, the offline SGD recursion will exhibit 'approximate' heavy tails. More precisely, as can be seen in (13), the tails will have a global power-law behavior due to the term $t^{-\alpha}$, where $\alpha$ is the tail index determined by online SGD. On the other hand, the power-law behavior is only approximate, as we have an additional term, which vanishes at a rate $\mathcal{O}(n^{-1/2})$. Hence, as $n$ increases, the power-law behavior will be more prominent, which might bring an explanation for why heavy tails are still observed even when $n$ is finite.

To establish Theorem 3, as an intermediate step, we show that $\pi^{(n)}$ converges to $\pi$ in the Wasserstein-1 metric. The result is given in the following theorem and can be interesting on its own.

---

[6]Since we are assuming Gaussian data, $\epsilon_n$ will converge to zero with an exponential rate. However, we do not focus on explicit calculations of this quantity as it is rather orthogonal to our problematic.

**Theorem 4.** *Under the setting of Theorem 3, for $p, q > 0$ with $1/p + 1/q = 1$, $p > d/2$, $q < \alpha$, the following holds with probability larger than $1 - \epsilon_n$:*

$$\mathbf{W}_1(\pi, \pi^{(n)}) \leq c_0 \frac{\eta}{1 - \delta_A} \mathbf{W}_p(\mu_{\mathbf{A},\mathbf{b}}, \mu_{\mathbf{A},\mathbf{b}}^{(n)}) \, , \tag{14}$$

*where $c_0 = (\mathbb{E}[\|X_0^{(n)}\|^q])^{1/q} + 1$ with $X_0^{(n)} \sim \pi^{(n)}$, and $\delta_A = \mathbb{E}[\|\, \mathrm{I}_d - \eta \mathbf{A}_1\|]$.*

This result shows that the stationary distribution of offline SGD will converge to the one of online SGD as $n \to \infty$, and the rate of this convergence is determined by how fast the empirical distribution of the dataset converges to the true data distribution, as measured by $\mathbf{W}_p(\mu_{\mathbf{A},\mathbf{b}}, \mu_{\mathbf{A},\mathbf{b}}^{(n)})$. Once (14) is established, to obtain Theorem 3, we use the relationship that $\mathbf{W}_1(\pi, \pi^{(n)}) \geq \mathbf{W}_1(\mathscr{L}(\|X_\infty^{(n)}\|), \mathscr{L}(\|X_\infty\|)) = \int_0^\infty |\mathbb{P}(\|X_\infty^{(n)}\| > t) - \mathbb{P}(\|X_\infty\| > t)| \, \mathrm{d}t$ (see Lemmas 1 and 3 in the Appendix). Finally, by exploiting the assumptions made on $\mu_{\mathbf{A},\mathbf{b}}$ in Theorem 1, we can invoke [FG15] (see Lemma 2 in Section A.1) and show that $\mathbf{W}_p(\mu_{\mathbf{A},\mathbf{b}}, \mu_{\mathbf{A},\mathbf{b}}^{(n)})$ behaves as $\mathcal{O}(n^{-1/2})$.

### 3.2 Strongly convex objectives

We now focus on more complex loss functions, as in Theorem 2. For simplicity, we consider $b = 1$. The general case, while similar, would complicate the notation and, in our opinion, make our results more difficult to digest. We start by rewriting offline SGD with respect to an empirical measure $\mu_z^{(n)}$:

$$X_{k+1}^{(n)} = X_k^{(n)} - \eta \nabla f(X_k^{(n)}, \hat{Z}_{k+1}^{(n)}) \, , \tag{15}$$

where $(\hat{Z}_k^{(n)})_{k \geq 1}$ are i.i.d. random variables associated with the empirical measure defined for any $\mathsf{A} \in \mathcal{Z}$ as:

$$\mu_z^{(n)}(\mathsf{A}) = n^{-1} \sum_{j=1}^n \delta_{\hat{Z}_j^{(n)}}(\mathsf{A}) \, . \tag{16}$$

We first make the following assumption.

**Assumption 1.** *(a) There exists $L > 0$ such that $\|\nabla f(x, z) - \nabla f(x, z')\| \leq L(\|x\| + 1)(\|z - z'\|)$, for any $z, z' \in \mathsf{Z}$ and $x \in \mathbb{R}^d$.*
*(b) The data distribution $\mu_z$ has finite $q$-th moments with some $q \geq 1$.*

Assumption 1-(a) is a Lipschitz-like condition that is useful for decoupling $x$ and $z$, and is commonly used in the analysis of optimization and stochastic approximations algorithms; see, e.g., [BWMP12]. We can now state our second main contribution, an extension of Theorem 2 to the offline setting.

**Theorem 5.** *Let $(X_k)_{k \geq 0}$ and $(X_k^{(n)})_{k \geq 0}$ be defined as in (2) and (4) with $b = 1$ and $n \geq 1$. Assume Assumption 1 and the conditions of Theorem 2 with $\beta > 1$ hold. Further assume that $R(Z_1)$ is non-deterministic (see Theorem 2), $(X_k)_{k \geq 0}$ is geometrically ergodic and $\epsilon_n \in [0, 1]$ is the probability that $(X_k^{(n)})_{k \geq 0}$ is not ergodic (in the Wasserstein sense). Then, for every $\varepsilon > 0$, there exist constants $c, c_\alpha, c_\beta$, and $t_0 \geq 0$, such that for all $t > t_0$, the following inequalities hold with probability larger than $1 - \epsilon_n$:*

$$\left[ \frac{1}{2^{\alpha + \epsilon}} \frac{c_\alpha}{t^{\alpha + \epsilon}} - \frac{c}{t} \mathbf{W}_1(\mu_z, \mu_z^{(n)}) \right] \leq \mathbb{P}(\|X_\infty^{(n)}\| > t) \leq \left[ 2^{\beta - \epsilon} \frac{c_\beta}{t^{\beta - \epsilon}} + \frac{2c}{t} \mathbf{W}_1(\mu_z, \mu_z^{(n)}) \right], \quad \forall n,$$

*where $c = (\mathbb{E}[\|X_0^{(n)}\|] + 1) L \eta / (1 - \delta_R)$ with $\delta_R = \mathbb{E}[R(Z_1)]$, and $X_0^{(n)} \sim \pi^{(n)}$.*

This theorem shows that the approximate power-laws will also be apparent for more general loss functions: for large enough $n$, the tails of offline SGD will be upper- and lower-bounded by two power-laws with exponents $\alpha$ and $\beta$. Note that the ergodicity of $(X_k)_{k \geq 0}$ can be established by using similar tools provided in [DDB20]. On the other hand, the assumption on $R(Z_1)$ being non-deterministic and $\mathbb{E}[R^\beta(Z_1)] = 1$ with $\beta > 1$ jointly indicate that $\delta_R = \mathbb{E}[R(Z_1)] < 1$, which is a form of contraction on average, hence the strong convexity condition is needed.

Similarly to Theorem 3, in order to prove this result, we first obtain a nonasymptotic Wasserstein convergence bound between $\pi^{(n)}$ and $\pi$, stated as follows:

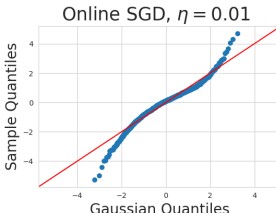
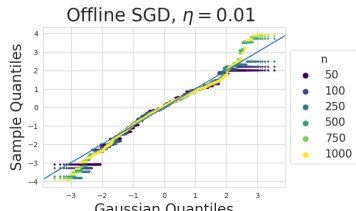

Figure 3: QQ-plots of 1D linear regression experiment. Left: Online SGD exhibits heavy-tails. Right: Offline SGD with varying $n$; larger $n$ exhibits heavier tails.

**Theorem 6.** *Under the setting of Theorem 5, the following holds with probability greater than $1 - \epsilon_n$:*

$$\mathbf{W}_1(\pi, \pi^{(n)}) \leq c_0 L \frac{\eta}{1 - \delta_R} \mathbf{W}_1(\mu_z, \mu_z^{(n)}) \,,$$

*with $c_0 = \mathbb{E}[\|X_0^{(n)}\|] + 1$, where $X_0^{(n)} \sim \pi^{(n)}$.*

The convergence rate implied by Theorem 6 is implicitly related to the finite moments of the data-generating distribution $\mu_z$. Using the results of [FG15] ( Lemma 2 in the Appendix), we can determine the convergence rate $r$, where with high probability it will hold that $\mathbf{W}_1(\pi, \pi^{(n)}) = \mathcal{O}(n^{-r})$, with $r \in (0, 1/2]$. Let the data have finite moments up to order $q$. Then, for example, if $q > 4d$, we get $r = 1/2$, i.e., the fastest rate. As another example: if $d > 2$ and $q = 3$, we obtain a rate $r = 1/d$. In other words, the moments of the data-generating distribution determine whether we can achieve a dimension-free rate or fall victim to the curse of dimensionality. Thus, if the data do not admit higher-order moments, our results show that we need a large number of data points $n$ to be able to observe power-law tails in offline SGD.

## 4    Experiments

In this section, we support our theoretical findings on both synthetic and real-world problems. Our primary objective is to validate the tail behavior of offline SGD iterates by adapting previous online SGD analyses [GSZ21, SSG19], in which the authors investigated how heavy-tailed behavior relates to the step size $\eta$ and batch size $b$, or their ratio $\eta/b$[7].

### 4.1    Linear regression

**Experimental setup.** In the linear regression examples, we examine the case with a quadratic loss and Gaussian data. In this simple scenario, it was observed that online SGD could exhibit heavy-tailed behavior [GSZ21]. The model can be sum-

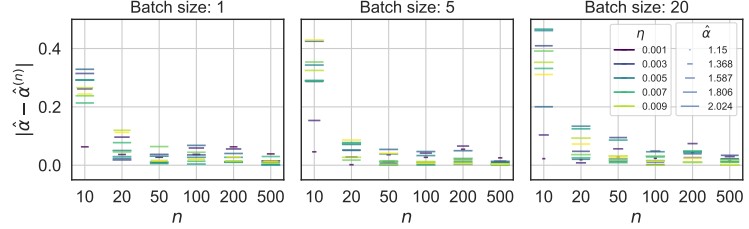

Figure 4: Estimated tail indices difference

marized as follows: with initial parameters $\bar{X}_0 \sim \mathrm{N}\left(0, \sigma_x^2 \, \mathrm{I}_d\right)$, features $a_i \sim \mathrm{N}\left(0, \sigma^2 \, \mathrm{I}_d\right)$, and targets $y_i \mid a_i, \bar{X}_0 \sim \mathrm{N}\left(a_i^\top \bar{X}_0, \sigma_y^2\right)$, for $i = 1, \ldots, n$, and $\sigma, \sigma_x, \sigma_y > 0$. In our experiments, we fix $\sigma = 1, \sigma_x = \sigma_y = 3$, use either $d = 1$ or $d = 100$, and simulate the statistical model to obtain $\{a_i, y_i\}_{i=1}^n$.

In order to examine the offline version of the model (i.e., for offline SGD), we utilize a finite dataset with $n$ points rather than observing new samples at each iteration. We then analyze the same experimental setting with an increasing number of samples $n$ and illustrate how similar heavy-tailed behavior emerges in offline SGD.

---

[7]The code scripts for reproducing the experimental results can be accessed at github.com/krunolp/offline_ht.

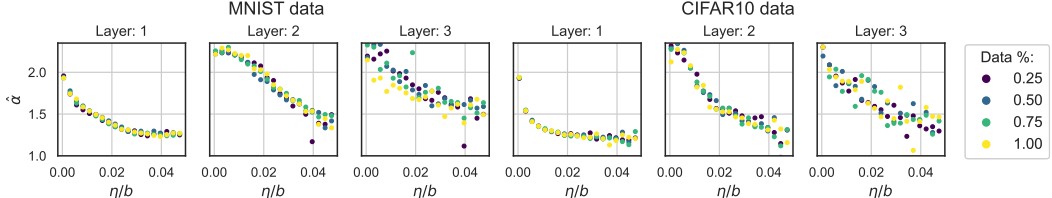

Figure 5: Estimated tail indices for a 3-layer fully connected NN

**Preliminary tail inspection.** We begin by analyzing how the tail behavior differs between online and offline SGD (with varying $n$ in the latter). Specifically, in Figure 3, we depict a 1-dimensional linear regression task by analyzing the QQ-plots of the estimated stabilizing distributions after 1000 iterations. We observe that online SGD with a sufficiently large $\eta$ exhibits heavy, non-Gaussian tails, underscoring the prevalence of heavy tails even in such simple scenarios. Furthermore, we also observe that offline SGD exhibits increasingly heavier tails as the sample size $n$ increases, as our theoretical results suggest.

**Tail estimation.** We now set $d = 100$ and run the corresponding offline and online SGD recursions. We then use a tail-index estimator [MMO15], which assumes that the recursions both converge to an *exact* heavy-tailed distribution. While this is true for online SGD due to Theorem 1, offline SGD will only possess *approximate* heavy-tails, and the power-law behavior might not be apparent for small $n$. Hence, for small $n$, we expect the estimated tail index for offline SGD will be inaccurate and get more and more accurate as we increase $n$.

We illustrate this in Figure 2, in which we plot the range of estimated offline SGD tail indices (marked in green) corresponding to 10 random initializations, while varying $n$ and $\eta$. We can see that, across all learning rates, the variance of the tail estimation decreases as $n$ increases and that the estimated values get closer to the estimated tail index for online SGD (marked as the horizontal blue line).

**Further analyses.** We now run online and offline SGD recursions, varying $\eta$ from 0.001 to 0.01, $b$ from 1 to 20, and $n$ from 1 to 500. Each hyperparameter configuration is run 1600 times with distinct initializations, yielding 1600 estimated samples from the stationary distributions, used to estimate the tail indexes. In order to estimate the tail-indexes $\alpha$ (online SGD) and $\alpha^{(n)}$ (offline SGD) of the respective stationary distributions, we follow the procedure as explained in [GSZ21]. Finally, in Figure 4, we plot the absolute difference of the estimated indexes, $|\hat{\alpha}^{(n)} - \hat{\alpha}|$.

We find that a larger number of data samples leads to a smaller discrepancy between the online and offline approximations across all batch sizes $b$ and step sizes $\eta$. This trend is consistently observed. Moreover, we observe that larger values of $\eta$ lead to a smaller discrepancy on average, across all $n$. The conclusion here is that, as $n$ increases, the power-law tails in offline SGD become more apparent and the estimator can identify the true tail index corresponding to online SGD even for moderately large $n$, which confirms our initial expectations.

### 4.2  Neural networks (NN)

To test the applicability of our theory in more practical scenarios, we conduct a second experiment using fully connected (FC) NNs (3 layers, 128 neurons, ReLU activations), as well as larger architectures, such as LeNet (60k parameters, 3 convolutional layers, 2 FC layers)[8][LBBH98], and AlexNet (62.3M parameters, 5 convolutional layers, 3 FC layers)[KSH17]. The models are trained for $10,000$ iterations using cross-entropy loss on the MNIST and CIFAR-10 datasets. We vary the learning rate from $10^{-4}$ to $10^{-1}$, and the batch size $b$ from 1 to 10, with offline SGD utilizing $25\%$, $50\%$, and $75\%$ of the training data.

We proceed similarly to the linear regression experiment and again replicate the method presented in [GSZ21]: we estimate the tail index per layer, and plot the corresponding results, with different colors representing different data proportions (1.00 indicates the full data set). The results for the fully connected network are presented in Figure 5 (on CIFAR-10 & MNIST), LeNet (on MNIST)

---

[8]For completeness, LeNet further uses 2 subsampling layers.

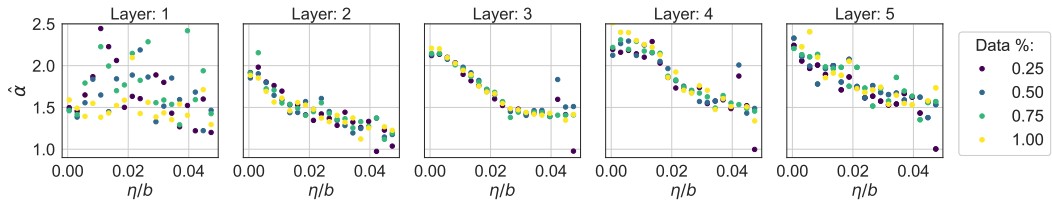

Figure 7: Estimated tail indices, LeNet, MNIST

in Figure 7, and AlexNet (on CIFAR-10) in Figure 6. The LeNet CIFAR-10 and AlexNet MNIST results can be found in the Appendix D.

Our observations show that the estimated $\hat{\alpha}^{(n)}$ has a strong correlation with $\hat{\alpha}$: using a reasonably large proportion of the data enables us to estimate the tail index that is measured over the whole dataset. Moreover, although the dependence of $\hat{\alpha}^{(n)}$ on $\eta/b$ varies between layers, the

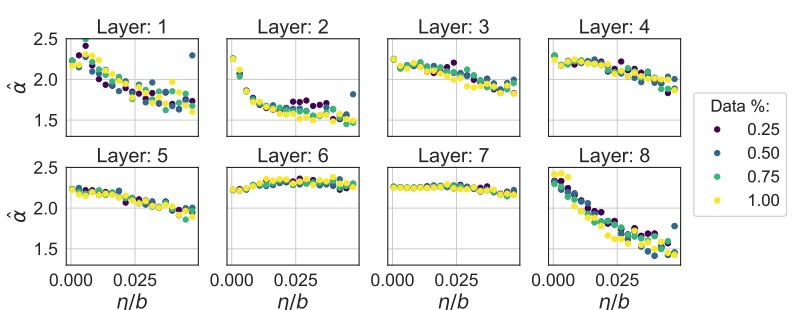

Figure 6: Estimated tail indices, AlexNet, CIFAR-10

measured $\hat{\alpha}^{(n)}$'s correlate well with the ratio $\eta/b$ across all datasets and NN architectures[9]. While our theory does not directly cover this setup, our results show that similar behavior is also observed in more complicated scenarios.

## 5 Conclusion

We established a relationship between the data-generating distributions and stationary distributions of offline and online SGD. This enabled us to develop the first theoretical result illuminating the heavy-tailed behavior in offline SGD. We extended previous results encompassing both quadratic losses, as well as more sophisticated strongly convex loss functions. Through an experimental study, we validated our theoretical findings in a variety of settings.

**Limitations.** There are two main limitations to our work. Firstly, our analysis focuses predominantly on the 'upper' tails of the distributions. Studying the bulk of the distribution of the iterates may reveal new findings. For example, research in this direction could connect our findings to learning theory by leveraging existing generalization bounds (e.g., see Corollary 2 in [HSKM22] for a link between 'lower' tails and generalization error). Secondly, it would be of great interest to extend our results to the non-convex deep learning settings. Finally, since this is a theoretical paper studying online and offline SGD, our work contains no direct potential negative societal impacts.

**Acknowledgments.** We thank Benjamin Dupuis for the valuable feedback. AD would like to thank the Isaac Newton Institute for Mathematical Sciences for support and hospitality during the programme *The mathematical and statistical foundation of future data-driven engineering* when work on this paper was undertaken. Umut Şimşekli's research is supported by the French government under management of Agence Nationale de la Recherche as part of the "Investissements d'avenir" program, reference ANR-19-P3IA-0001 (PRAIRIE 3IA Institute) and the European Research Council Starting Grant DYNASTY – 101039676.

---

[9]The monotonic relation between the tail exponent and the $\eta/b$ ratio is in line with findings from [GSZ21], although new findings (see Sec.6.6 in [ZLMU22]) point out that a mere dependence on this ratio has been found broken. We believe that for other ranges of the parameters (outside of the ones originally used in [GSZ21]), a different behavior could be observed.

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

# Approximate Heavy Tails in Offline (Multi-Pass) Stochastic Gradient Descent

## APPENDIX

The organization of the appendix is as follows:

- In Section A, we provide the background material required for the proof methodology.
- In Section B, we present technical results used to conclude the results of our work.
- In Section C, we give the proofs of Theorems 3-6.
- In Section D, we provide additional experimental results.
- In Section E, we justify the tail estimator choice.

## A  Additional Technical Background

### A.1  Existing results

**Lemma 1** (Explicit solution of Wasserstein distance [PZ19])**.** *Let $\mu_1, \mu_2 \in \mathcal{P}(\mathbb{R})$ be two probability measures on $\mathbb{R}$, and denote their cumulative distribution functions by $F_1(x)$ and $F_2(x)$ respectively. Then, the Wasserstein-$p$ distance between $\mu_1$ and $\mu_2$ has an explicit formula:*

$$\mathbf{W}_p\left(\mu_1, \mu_2\right) = \left(\int_0^1 \left|F_1^{-1}(q) - F_2^{-1}(q)\right|^p \, \mathrm{d}q\right)^{1/p},$$

*where $F_1^{-1}$ and $F_2^{-1}$ denote the quantile functions. In the case when $p = 1$, by applying the change of variables, one can obtain the following:*

$$\mathbf{W}_1\left(\mu_1, \mu_2\right) = \int_{\mathbb{R}} |F_1(x) - F_2(x)| \, \mathrm{d}x.$$

Now, for $q > 0, \alpha > 0, \gamma > 0$ and $\mu \in \mathcal{P}\left(\mathbb{R}^d\right)$, let:

$$M_q(\mu) := \int_{\mathbb{R}^d} |x|^q \mu(\mathrm{d}x) \text{ and } \mathcal{E}_{\alpha,\gamma}(\mu) := \int_{\mathbb{R}^d} \mathrm{e}^{\gamma|x|^\alpha} \mu(\mathrm{d}x).$$

Furthermore, consider an i.i.d. sequence $(X_k)_{k \geq 1}$ of $\mu$-distributed random variables and, for $n \geq 1$, define the empirical measure by:

$$\mu^{(n)} := \frac{1}{n} \sum_{k=1}^n \delta_{X_k}.$$

We can now proceed to state the following result.

**Lemma 2** ([FG15])**.** *Let $\mu \in \mathcal{P}\left(\mathbb{R}^d\right)$ and let $p > 0$. Assume one of the three following conditions:*
*(1) $\alpha > p, \gamma > 0, \mathcal{E}_{\alpha,\gamma}(\mu) < \infty$,*
*(2) $\alpha \in (0, p), \gamma > 0, \mathcal{E}_{\alpha,\gamma}(\mu) < \infty$,*
*(3) $q > 2p, M_q(\mu) < \infty$.*
*Then for all $n \geq 1$, all $x \in (0, \infty)$,*

$$\mathbb{P}\left(\mathbf{W}_p(\mu^{(n)}, \mu) \geq x\right) \leq a(n, x)\mathbf{1}_{\{x \leq 1\}} + b(n, x)$$

*where*

$$a(n, x) = C \begin{cases} \exp\left(-cnx^2\right) & \text{if } p > d/2 \\ \exp\left(-cn(x/\log(2 + 1/x))^2\right) & \text{if } p = d/2 \\ \exp\left(-cnx^{d/p}\right) & \text{if } p \in [1, d/2) \end{cases}$$

*and*

$$b(n, x) = C \begin{cases} \exp\left(-cnx^{\alpha/p}\right) \mathbf{1}_{\{x>1\}} & \text{under (1),} \\ \exp\left(-c(nx)^{(\alpha-\varepsilon)/p}\right) \mathbf{1}_{\{x\leq1\}} + \exp\left(-c(nx)^{\alpha/p}\right) \mathbf{1}_{\{x>1\}} & \forall \varepsilon \in (0, \alpha) \text{ under (2),} \\ n(nx)^{-(q-\varepsilon)/p} & \forall \varepsilon \in (0, q) \text{ under (3).} \end{cases}$$

*The positive constants $C$ and $c$ depend only on $p, d$ and either on $\alpha, \gamma, \mathcal{E}_{\alpha,\gamma}(\mu)$ (under (1)) or on $\alpha, \gamma, \mathcal{E}_{\alpha,\gamma}(\mu), \varepsilon$ (under (2)) or on $q, M_q(\mu), \varepsilon$ (under (3)).*

# B  Technical Lemmas

**Lemma 3.** *Let $X$ and $Y$ be two random vectors in $\mathbb{R}^d$. Then, we have*

$$\mathbf{W}_1(\mathscr{L}(X), \mathscr{L}(Y)) \geq \mathbf{W}_1(\mathscr{L}(\|X\|), \mathscr{L}(\|Y\|)). \tag{17}$$

*Proof.* By the definition of the Wasserstein distance, we have that

$$\mathbf{W}_1(\mathscr{L}(X), \mathscr{L}(Y)) = \inf_{\gamma \in \Gamma(\mathscr{L}(X), \mathscr{L}(Y))} \int_{\mathbb{R}^d \times \mathbb{R}^d} \|x - y\| \mathrm{d}\gamma(x, y) \tag{18}$$

$$= \int_{\mathbb{R}^d \times \mathbb{R}^d} \|x - y\| \mathrm{d}\gamma^*(x, y), \tag{19}$$

where $\gamma^*$ is the coupling that attains the infimum (the proof of the existence of such coupling for any Wasserstein-$p$ distance, where $p \geq 1$, can be found in, e.g., [Bog06, Theorem 8.10.45]). Then, by the reverse triangle inequality, we have

$$\mathbf{W}_1(\mathscr{L}(X), \mathscr{L}(Y)) \geq \int_{\mathbb{R}^d \times \mathbb{R}^d} \left| \|x\| - \|y\| \right| \mathrm{d}\gamma^*(x, y) \tag{20}$$

$$= \int_{\mathbb{R}_+ \times \mathbb{R}_+} |x - y| \, \mathrm{d}T_\# \gamma^*(x, y), \tag{21}$$

where $T : \mathbb{R}^d \times \mathbb{R}^d \mapsto \mathbb{R}_+ \times \mathbb{R}_+$ is the map $(x, y) \mapsto (\|x\|, \|y\|)$ and $T_\# \gamma^*$ is the pushforward measure such that for a measurable set $B \subset \mathbb{R}_+ \times \mathbb{R}_+$, we have $T_\# \gamma^*(B) = \gamma^*(T^{-1}(B))$. As $T_\# \gamma^*$ is a coupling between $\|X\|$ and $\|Y\|$, we have that

$$\mathbf{W}_1(\mathscr{L}(X), \mathscr{L}(Y)) \geq \inf_{\gamma \in \Gamma(\mathscr{L}(\|X\|), \mathscr{L}(\|Y\|))} \int_{\mathbb{R}_+ \times \mathbb{R}_+} |x - y| \, \mathrm{d}\gamma(x, y) \tag{22}$$

$$= \mathbf{W}_1(\mathscr{L}(\|X\|), \mathscr{L}(\|Y\|)). \tag{23}$$

This concludes the proof. $\qquad \square$

# C  Proofs

This section contains the proofs of our theoretical findings.

## C.1  Proof of Theorem 4

*Proof.* First, note that the unique stationary distributions of $(X_k)_{k \geq 0}$ and $(X_k^{(n)})_{k \geq 0}$ are denoted respectively by $\pi$ and $\pi^{(n)}$. Denote by $E_n$ the event on which $(X_k^{(n)})_{k \geq 0}$ has a stationary distribution $\pi^{(n)}$ such that $\mathbb{P}(E_n) \geq 1 - \epsilon_n$ (where $\epsilon_n \in [0, 1]$ is the probability that $(X_k^{(n)})_{k \geq 0}$ is not ergodic, in the Wasserstein sense). Given $E_n$, let $(\mathbf{A}_k, \mathbf{A}_k^{(n)})_{k \geq 1}$ and $(\mathbf{b}_k, \mathbf{b}_k^{(n)})_{k \geq 1}$ be the sequences of i.i.d. optimal couplings for $\mu_{\mathbf{A}}$ and $\mu_{\mathbf{A}}^{(n)}$, $\mu_{\mathbf{b}}$ and $\mu_{\mathbf{b}}^{(n)}$ respectively. Therefore, by construction, for any $k \in \mathbb{N}$, $\mathbb{E}[\|\mathbf{A}_k - \mathbf{A}_k^{(n)}\|] = \mathbf{W}_1(\mu_{\mathbf{A}}, \mu_{\mathbf{A}}^{(n)})$, and $\mathbb{E}[\|\mathbf{b}_k - \mathbf{b}_k^{(n)}\|] = \mathbf{W}_1(\mu_{\mathbf{b}}, \mu_{\mathbf{b}}^{(n)})$. Based on these two sequences, we consider the processes $(X_k)_{k \geq 0}$, $(X_k^{(n)})_{k \geq 0}$, $(Y_k)_{k \geq 0}$ defined by the recursions:

1. $X_{k+1} = (\mathbf{I}_d - \eta \mathbf{A}_{k+1}) X_k + \eta \mathbf{b}_{k+1}$ , where $X_0 \sim \pi$,

2. $X_{k+1}^{(n)} = (\mathbf{I}_d - \eta \mathbf{A}_{k+1}^{(n)}) X_k^{(n)} + \eta \mathbf{b}_{k+1}^{(n)}$, where $X_0^{(n)} \sim \pi^{(n)}$,

3. $Y_{k+1} = (\mathbf{I}_d - \eta \mathbf{A}_{k+1}) Y_k + \eta \mathbf{b}_{k+1}$, where $Y_0 = X_0^{(n)}$.

Note that $(X_k)_{k \geq 0}$ corresponds to the online SGD recursion (7), and $(X_k^{(n)})_{k \geq 0}$ to the offline SGD recursion (9). In addition, since these two Markov chains are started at stationarity, we have:

$$\mathbf{W}_1(\pi, \pi^{(n)}) = \mathbf{W}_1(\mathscr{L}(X_0), \mathscr{L}(X_0^{(n)})) = \mathbf{W}_1(\mathscr{L}(X_k), \mathscr{L}(X_k^{(n)})), \ \forall k.$$

With these definitions at hand and this observation, using triangle inequality, we can obtain:

$$\mathbf{W}_1(\pi, \pi^{(n)}) = \mathbf{W}_1(\mathscr{L}(X_k), \mathscr{L}(X_k^{(n)}))$$
$$\leq \mathbf{W}_1(\mathscr{L}(X_k), \mathscr{L}(Y_k)) + \mathbf{W}_1(\mathscr{L}(Y_k), \mathscr{L}(X_k^{(n)})).$$

Now, by [GSZ21, Theorem 8], we have that $Y_k$ is geometrically ergodic with respect to its stationary distribution $\pi$, i.e., there exist constants $c_\rho > 0$, $\rho \in (0, 1)$ such that the following inequality holds:

$$\mathbf{W}_1(\mathscr{L}(Y_k), \pi) \leq c_\rho \mathbf{W}_1(\mathscr{L}(Y_0), \pi) e^{-\rho k} \text{ , for any } k \geq 0 . \tag{24}$$

Therefore, we have:

$$\mathbf{W}_1(\pi, \pi^{(n)}) \leq c_\rho e^{-\rho k} \mathbf{W}_1(\pi, \mathscr{L}(Y_0)) + \mathbf{W}_1(\mathscr{L}(Y_k), \mathscr{L}(X_k^{(n)})),$$

as $\mathbf{W}_1(\mathscr{L}(X_k), \mathscr{L}(Y_k)) = \mathbf{W}_1(\pi, \mathscr{L}(Y_k)) \leq c_\rho e^{-\rho k} \mathbf{W}_1(\pi, \mathscr{L}(Y_0))$. Rearranging the above terms, we get $\mathbf{W}_1(\pi, \pi^{(n)})(1 - c_\rho e^{-\rho k}) \leq \mathbf{W}_1(\mathscr{L}(Y_k), \mathscr{L}(X_k^{(n)}))$ implying:

$$\mathbf{W}_1(\pi, \pi^{(n)}) \leq (1 - c_\rho e^{-\rho k})^{-1} \mathbf{W}_1(\mathscr{L}(Y_k), \mathscr{L}(X_k^{(n)})). \tag{25}$$

To bound the right-hand side of (25), we consider the following difference by using the recursion definitions:

$$Y_{k+1} - X_{k+1}^{(n)} = Y_k - \eta \mathbf{A}_{k+1} Y_k + \eta \mathbf{b}_{k+1} - X_k^{(n)} + \eta \mathbf{A}_{k+1}^{(n)} X_k^{(n)} - \eta \mathbf{b}_{k+1}^{(n)} \tag{26}$$
$$= (\mathrm{I}_d - \eta \mathbf{A}_{k+1})(Y_k - X_k^{(n)}) - \eta (\mathbf{A}_{k+1} - \mathbf{A}_{k+1}^{(n)}) X_k^{(n)} + \eta (\mathbf{b}_{k+1} - \mathbf{b}_{k+1}^{(n)}). \tag{27}$$

Writing out the recursion, we can obtain:

$$Y_{k+1} - X_{k+1}^{(n)} = (Y_0 - X_0^{(n)}) \prod_{i=0}^{k} (\mathrm{I}_d - \eta \mathbf{A}_{i+1})$$
$$- \eta \sum_{i=0}^{k} \left[ X_i^{(n)} (\mathbf{A}_{i+1} - \mathbf{A}_{i+1}^{(n)}) \prod_{j=i+2}^{k+1} (\mathrm{I}_d - \eta \mathbf{A}_j) \right]$$
$$+ \eta \sum_{i=0}^{k} \left[ (\mathbf{b}_{i+1} - \mathbf{b}_{i+1}^{(n)}) \prod_{j=i+2}^{k+1} (\mathrm{I}_d - \eta \mathbf{A}_j) \right],$$

where for any sequence $(a_i)_{i \geq 0}$, we let $\prod_{i=j}^{k} a_i = 1$ when $j > k$. Now, taking the norm of both sides, using the triangle inequality, and taking expectations given $E_n$, we obtain:

$$\mathbb{E}[\|Y_{k+1} - X_{k+1}^{(n)}\|] \leq \mathbb{E}[\|(Y_0 - X_0^{(n)}) \prod_{i=0}^{k} (\mathrm{I}_d - \eta \mathbf{A}_{i+1})\|]$$
$$+ \eta \mathbb{E}[\sum_{i=0}^{k} \|X_i^{(n)} (\mathbf{A}_{i+1} - \mathbf{A}_{i+1}^{(n)}) \prod_{j=i+2}^{k+1} (\mathrm{I}_d - \eta \mathbf{A}_j)\|] \tag{28}$$
$$+ \eta \mathbb{E}[\sum_{i=0}^{k} \|(\mathbf{b}_{i+1} - \mathbf{b}_{i+1}^{(n)}) \prod_{j=i+2}^{k+1} (\mathrm{I}_d - \eta \mathbf{A}_j)\|].$$

We shall now analyze each of the three summands separately. For the first term in (28), as we start with $Y_0 = X_0^{(n)}$, it equals zero. For the second term in (28), we have:

$$\eta\mathbb{E}[\sum_{i=0}^{k} \|X_i^{(n)}(\mathbf{A}_{i+1} - \mathbf{A}_{i+1}^{(n)}) \prod_{j=i+2}^{k+1} (\mathrm{I}_d - \eta\mathbf{A}_j)\|]$$

$$\overset{(a)}{=} \eta \sum_{i=0}^{k} \mathbb{E}[\|X_i^{(n)}(\mathbf{A}_{i+1} - \mathbf{A}_{i+1}^{(n)}) \prod_{j=i+2}^{k+1} (\mathrm{I}_d - \eta\mathbf{A}_j)\|]$$

$$\overset{(b)}{\leq} \eta \sum_{i=0}^{k} \mathbb{E}[\|X_i^{(n)}(\mathbf{A}_{i+1} - \mathbf{A}_{i+1}^{(n)})\|]\mathbb{E}^{k-i}[\|\mathrm{I}_d - \eta\mathbf{A}_1\|]$$

$$\overset{(c)}{\leq} \eta \sum_{i=0}^{k} (\mathbb{E}[\|X_i^{(n)}\|^q])^{\frac{1}{q}} (\mathbb{E}[\|\mathbf{A}_{i+1} - \mathbf{A}_{i+1}^{(n)}\|^p])^{\frac{1}{p}} \mathbb{E}^{k-i}[\|\mathrm{I}_d - \eta\mathbf{A}_1\|],$$

where (a) follows from the linearity of expectation, (b) follows as $\mathbf{A}_1, \ldots, \mathbf{A}_{i-1}$ are i.i.d. and independent from $\mathbf{A}_i, \ldots, \mathbf{A}_{k+1}$, and (c) is obtained by applying the Hölder's inequality, where $p$ and $q$ are such that $1/p + 1/q = 1$, with $p, q \in (1, +\infty)$, $q < \alpha$ and $p > d/2$ (used later in the proof), where $\alpha$ is the tail index.

As the chain $(X_k^{(n)})_{k\geq 0}$ starts from its stationary distribution $\pi^{(n)}$, we have that $(\mathbb{E}[\|X_i^{(n)}\|^q])^{1/q} = (\mathbb{E}[\|X_0^{(n)}\|^q])^{1/q}, \forall i$. Using the fact that $(\mathbf{A}_i - \mathbf{A}_i^{(n)})_{i=1}^{k+1}$ are i.i.d. random variables (the randomness arises from the empirical measure $\mu_\mathbf{A}^{(n)}$), $\mathbb{E}[\|\mathrm{I}_d - \eta\mathbf{A}_1\|^\alpha] = 1$ and $\alpha > 1$, we have that:

$$\delta_A := \mathbb{E}[\|\mathrm{I}_d - \eta\mathbf{A}_1\|] < 1. \tag{29}$$

Therefore, we can obtain the following bound:

$$\eta\mathbb{E}[\sum_{i=0}^{k} \|X_i^{(n)}(\mathbf{A}_{i+1} - \mathbf{A}_{i+1}^{(n)}) \prod_{j=i+2}^{k+1} (\mathrm{I}_d - \eta\mathbf{A}_j)\|]$$

$$\leq \eta(\mathbb{E}[\|X_0^{(n)}\|^q])^{\frac{1}{q}} (\mathbb{E}[\|\mathbf{A}_1 - \mathbf{A}_1^{(n)}\|^p])^{\frac{1}{p}} \sum_{i=0}^{k} \mathbb{E}^{k-i}[\|\mathrm{I}_d - \eta\mathbf{A}_1\|]$$

$$\leq \eta(\mathbb{E}[\|X_0^{(n)}\|^q])^{\frac{1}{q}} (\mathbb{E}[\|\mathbf{A}_1 - \mathbf{A}_1^{(n)}\|^p])^{\frac{1}{p}} \frac{1}{1 - \delta_A},$$

where the last equality follows from (29) and the power series sum formula. Using the same arguments, we can bound the third term in (28):

$$\eta\mathbb{E}[\sum_{i=0}^{k} \|(\mathbf{b}_{i+1} - \mathbf{b}_{i+1}^{(n)}) \prod_{j=i+2}^{k+1} (\mathrm{I}_d - \eta\mathbf{A}_j)\|] \leq \eta\mathbb{E}[\|\mathbf{b}_1 - \mathbf{b}_1^{(n)}\|] \frac{1}{1 - \delta_A}.$$

Combining the above, we can obtain that:

$$\mathbb{E}[\|Y_{k+1} - X_{k+1}^{(n)}\|] \leq \eta(\mathbb{E}[\|X_0^{(n)}\|^q])^{\frac{1}{q}} (\mathbb{E}[\|\mathbf{A}_1 - \mathbf{A}_1^{(n)}\|^p])^{\frac{1}{p}} \frac{1}{1 - \delta_A} + \eta\mathbb{E}[\|\mathbf{b}_1 - \mathbf{b}_1^{(n)}\|]\frac{1}{1 - \delta_A}$$

$$= \frac{\eta}{1 - \delta_A} \left( (\mathbb{E}[\|X_0^{(n)}\|^q])^{\frac{1}{q}} (\mathbb{E}[\|\mathbf{A}_1 - \mathbf{A}_1^{(n)}\|^p])^{\frac{1}{p}} + \mathbb{E}[\|\mathbf{b}_1 - \mathbf{b}_1^{(n)}\|] \right)$$

$$\leq \frac{\eta}{1 - \delta_A} \left( (\mathbb{E}[\|X_0^{(n)}\|^q])^{\frac{1}{q}} (\mathbb{E}[\|\mathbf{A}_1 - \mathbf{A}_1^{(n)}\|^p])^{\frac{1}{p}} + (\mathbb{E}[\|\mathbf{b}_1 - \mathbf{b}_1^{(n)}\|^p])^{\frac{1}{p}} \right)$$

$$= \frac{\eta}{1 - \delta_A} \left( (\mathbb{E}[\|X_0^{(n)}\|^q])^{\frac{1}{q}} \mathbf{W}_p(\mu_\mathbf{A}, \mu_\mathbf{A}^{(n)}) + \mathbf{W}_p(\mu_\mathbf{b}, \mu_\mathbf{b}^{(n)}) \right) \tag{30}$$

$$\leq \frac{\eta}{1 - \delta_A} \left( ((\mathbb{E}[\|X_0^{(n)}\|^q])^{\frac{1}{q}} + 1)\mathbf{W}_p(\mu_{\mathbf{A},\mathbf{b}}, \mu_{\mathbf{A},\mathbf{b}}^{(n)}) \right), \tag{31}$$

where the penultimate inequality follows from Jensen's inequality: it implies that $\mathbf{W}_p(\mu, \nu) \leq \mathbf{W}_q(\mu, \nu)$, for $p \leq q$ (see, e.g., [Vil09]). The last inequality follows as $\mathbf{W}_p(\mu_\mathbf{A}, \mu_\mathbf{A}^{(n)}) \leq \mathbf{W}_p(\mu_{\mathbf{A},\mathbf{b}}, \mu_{\mathbf{A},\mathbf{b}}^{(n)})$, and $\mathbf{W}_p(\mu_\mathbf{b}, \mu_\mathbf{b}^{(n)}) \leq \mathbf{W}_p(\mu_{\mathbf{A},\mathbf{b}}, \mu_{\mathbf{A},\mathbf{b}}^{(n)})$. Together with (25), we have:

$$\mathbf{W}_1(\pi, \pi^{(n)}) \leq ((\mathbb{E}[\|X_0^{(n)}\|^q])^{\frac{1}{q}} + 1)(1 - c_\rho e^{-\rho k})^{-1} \frac{\eta}{1 - \delta_A} \mathbf{W}_p(\mu_{\mathbf{A},\mathbf{b}}, \mu_{\mathbf{A},\mathbf{b}}^{(n)}) \ .$$

By taking the limit as $k \to \infty$, we can conclude the proof of Theorem 4. $\qquad\square$

### C.2 Proof of Theorem 3

*Proof.* Using results from Theorem 4, we now proceed to bound the Wasserstein-$p$ distance between the probability laws of $\mathbf{A}_1$ and $\mathbf{A}_1^{(n)}$. Reminiding ourselves that each element in the i.i.d. sequence $(a_k)_{k \geq 1}$ follows a $\mathrm{N}(0, \sigma^2 \mathrm{I}_d)$, we denote this measure with $\mu_a$. Correspondingly, we denote its empirical measure with $\mu_a^{(n)} = n^{-1} \sum_{j=1}^n \delta_{a_j}$. Given $E_n$, we let $(a_k, a_k^{(n)})_{k \geq 1}$ be the sequence of i.i.d. optimal couplings for $\mu_a$ and $\mu_a^{(n)}$, so that by construction, for any $k \in \mathbb{N}$, $(\mathbb{E}[\|a_k - a_k^{(n)}\|^{2p}])^{1/2p} = \mathbf{W}_{2p}(\mu_a, \mu_a^{(n)})$. Now:

$$\mathbf{W}_p(\mu_{\mathbf{A}}, \mu_{\mathbf{A}}^{(n)}) \leq (\mathbb{E}[\|\frac{1}{b} \sum_{i \in \Omega_1} a_i a_i^\top - \frac{1}{b} \sum_{j \in \Omega_1^{(n)}} a_j a_j^\top \|^p])^{1/p}$$

$$\overset{(a)}{\leq} \frac{1}{b} b (\mathbb{E}[\|(a_1 a_1^\top - a_1^{(n)} a_1^{(n)\top})\|^p])^{1/p}$$

$$= (\mathbb{E}[\|a_1 a_1^\top - a_1 a_1^{(n)\top} + a_1 a_1^{(n)\top} - a_1^{(n)} a_1^{(n)\top}\|^p])^{1/p}$$

$$\overset{(b)}{\leq} (\mathbb{E}[\|a_1 a_1^\top - a_1 a_1^{(n)\top}\|^p])^{1/p} + (\mathbb{E}[\|a_1 a_1^{(n)\top} - a_1^{(n)} a_1^{(n)\top}\|^p])^{1/p}$$

$$= (\mathbb{E}[\|a_1\|^p \|a_1 - a_1^{(n)}\|^p])^{1/p} + (\mathbb{E}[\|a_1^{(n)}\|^p \|a_1 - a_1^{(n)}\|^p])^{1/p}$$

$$\overset{(c)}{\leq} \left( (\mathbb{E}[\|a_1\|^{2p}])^{1/2p} + (\mathbb{E}[\|a_1^{(n)}\|^{2p}])^{1/2p} \right) (\mathbb{E}[\|a_1 - a_1^{(n)}\|^{2p}])^{1/2p}$$

where (a) follows from Minkowski's inequality and the fact that $(a_k)_{k \geq 1}$ are i.i.d. random variables, as well as $(a_k^{(n)})_{k \geq 1}$. Inequality (b) follows due to Minkowski's inequality, and (c) follows using the generalized Hölder's inequality with $1/2p + 1/2p = 1/p$.

Now, we can proceed to utilize Lemma 2 and (30). In order to apply Lemma 2 to the difference of measures between $\mathbf{A}_1$ and $\mathbf{A}_1^{(n)}$ (i.e., between $a_1$ and $a_1^{(n)}$), as $\mathbb{E}[\|X_0^{(n)}\|^q]^{1/q} < \infty$ for all $q < \alpha$, we require $q < \alpha$ and $1/p + 1/q = 1$ (due to Hölder's inequality).

First, we select $p$ large and $q$ small enough for both $q < \alpha$ and $p > d/2$. Now, in order to satisfy the requirements of Lemma 2, we set $\epsilon^* := \sqrt{\frac{1}{nC_1} \log \frac{4c_1}{\zeta}}$. This choice allows us to obtain that, with probability greater than $1 - \zeta/2$, $(\mathbb{E}[\|a_1 - a_1^{(n)}\|^{2p}])^{1/2p} = \mathbf{W}_{2p}(\mu_a, \mu_a^{(n)}) < \epsilon^*$. Therefore, we have that, with probability greater than $1 - \zeta/2$:

$$(\mathbb{E}[\|a_1 - a_1^{(n)}\|^{2p}])^{1/2p} \leq \sqrt{\frac{1}{nC_1} \log \frac{4c_1}{\zeta}} = \mathcal{O}(n^{-1/2}),$$

where the positive constants $C_1$ and $c_1$ depend only on $p$, $d$, $\mu_{a_1}$. Therefore, with probability greater than $1 - \zeta/2$:

$$(\mathbb{E}[\|\mathbf{A}_1 - \mathbf{A}_1^{(n)}\|^p])^{1/p} \leq ((\mathbb{E}[\|a_1\|^{2p}])^{1/2p} + (\mathbb{E}[\|a_1^{(n)}\|^{2p}])^{1/2p}) \sqrt{\frac{1}{nC_1} \log \frac{4c_1}{\zeta}}$$

$$= C_2 n^{-1/2} \sqrt{\log \frac{c_3}{\zeta}}, \tag{32}$$

with $C_2 = \sqrt{\frac{1}{C_1}} ((\mathbb{E}[\|a_1\|^{2p}])^{1/2p} + (\mathbb{E}[\|a_1^{(n)}\|^{2p}])^{1/2p})$ and $c_3 = 4c_1$.

In order to bound $(\mathbb{E}[\|\mathbf{b}_k - \mathbf{b}_k^{(n)}\|^p])^{1/p}$, we can apply Lemma 2 again to obtain constants $C_4$ and $c_5$ such that, with probability greater than $1 - \zeta/2$ :

$$(\mathbb{E}[\|\mathbf{b}_k - \mathbf{b}_k^{(n)}\|^p])^{1/p} \leq C_4 n^{-1/2} \sqrt{\log \frac{c_5}{\zeta}} = \mathcal{O}(n^{-1/2}). \tag{33}$$

Using (32), (33) with Theorem 4 (see (25) and (30)), and denoting $C_6 = C_2(\mathbb{E}[\|X_0^{(n)}\|^q])^{1/q} + C_4$ and $c_7 = \max\{c_3, c_5\}$ we can obtain, with probability greater than $1 - \zeta$:

$$\mathbf{W}_1(\pi, \pi^{(n)}) \leq \frac{\eta}{1 - \delta}\left(C_6\sqrt{\log\frac{c_7}{\zeta}}\right)n^{-1/2}. \tag{34}$$

Therefore, we obtain that, with probability greater than $1 - \zeta$, $\mathbf{W}_1(\pi, \pi^{(n)}) = \mathcal{O}(n^{-1/2})$.

Due to [GSZ21] we have $\lim_{t \to \infty} t^\alpha \mathbb{P}(\|X_\infty\| > t) \in (0, \infty)$. Now, let $\epsilon > 0$. Then, there exists $t_0$ s.t. for all $t \geq t_0$, $|t^\alpha \mathbb{P}(\|X_\infty\| > t) - c| \leq \epsilon$, for some $c \in (0, \infty)$. From (34), we know that with probability greater than $1 - \zeta$, there exist constants $\tilde{c}_1, \tilde{c}_2$ such that $\mathbf{W}_1(\pi, \pi^{(n)}) \leq \tilde{c}_1\sqrt{\log\frac{\tilde{c}_2}{\zeta}}n^{-1/2}$.

Now, using Lemma 3, we can obtain that, with probability greater than $1 - \epsilon_n - \zeta$:

$$\tilde{c}_1\sqrt{\log\frac{\tilde{c}_2}{\zeta}}\frac{1}{n^{1/2}} \geq \mathbf{W}_1(\pi, \pi^{(n)})$$

$$\geq \int_0^\infty |\mathbb{P}(\|X_\infty\| > t) - \mathbb{P}(\|X_\infty^{(n)}\| > t)| \, dt$$

$$\geq \int_{t'}^{t''} |\mathbb{P}(\|X_\infty\| > t) - \mathbb{P}(\|X_\infty^{(n)}\| > t)| \, dt,$$

where we have used that $\lim_{t \to \infty} \mathbb{P}(\|X_\infty\| > t) = 0$ and $\lim_{t \to \infty} \mathbb{P}(\|X_\infty^{(n)}\| > t) = 0$, in order to select $t'$ and $t''$ large enough for the last inequality to hold. Now, we have that

$$\int_{t'}^{t''} |\mathbb{P}(\|X_\infty\| > t) - \mathbb{P}(\|X_\infty^{(n)}\| > t)| \, dt \geq \int_{t'}^{t''} \frac{c - \epsilon}{t^\alpha} \, dt - \int_{t'}^{t''} \mathbb{P}(\|X_\infty^{(n)}\| > t) \, dt$$

$$\geq (t'' - t')\frac{(c - \epsilon)}{(t'')^\alpha} - (t'' - t')\mathbb{P}(\|X_\infty^{(n)}\| > t').$$

Therefore, by choosing $t'' = 2t'$, we can obtain:

$$\mathbb{P}(\|X_\infty^{(n)}\| > t') \geq \frac{c - \epsilon}{2^\alpha(t')^\alpha} - \frac{\tilde{c}_1}{t'n^{1/2}}\sqrt{\log\frac{\tilde{c}_2}{\zeta}}. \tag{35}$$

Similarly, for any $\epsilon > 0$ and $t > t_0$, we have $t^\alpha \mathbb{P}(\|X_\infty\| > t) \leq \epsilon + c$, and using similar arguments we obtain:

$$\tilde{c}_1\sqrt{\log\frac{\tilde{c}_2}{\zeta}}\frac{1}{n^{1/2}} \geq \int_{t'}^{t''} \mathbb{P}(\|X_\infty^{(n)}\| > t) \, dt - \int_{t'}^{t''} \frac{c + \epsilon}{t^\alpha} \, dt$$

$$\geq (t'' - t')\mathbb{P}(\|X_\infty^{(n)}\| > t'') - (t'' - t')\frac{c + \epsilon}{(t')^\alpha}.$$

Finally, choosing as before $t'' = 2t'$, we can obtain:

$$\mathbb{P}(\|X_\infty^{(n)}\| > 2t') \leq \frac{c + \epsilon}{(t')^\alpha} + \frac{\tilde{c}_1}{t'n^{1/2}}\sqrt{\log\frac{\tilde{c}_2}{\zeta}}. \tag{36}$$

Substituting for $\bar{t} = 2t'$, we can obtain:

$$\mathbb{P}(\|X_\infty^{(n)}\| > \bar{t}) \leq \frac{2^\alpha(c + \epsilon)}{\bar{t}^\alpha} + \frac{2\tilde{c}_1}{\bar{t}n^{1/2}}\sqrt{\log\frac{\tilde{c}_2}{\zeta}}. \tag{37}$$

Combining (35) and (37), we obtain that with probability greater than $1 - \zeta - \epsilon_n$, for any $\epsilon > 0$ and $t > t_0$:

$$\mathbb{P}(\|X_\infty^{(n)}\| > t) \geq \frac{1}{2^\alpha}\frac{c - \epsilon}{t^\alpha} - \frac{\tilde{c}_1}{tn^{1/2}}\sqrt{\log\frac{\tilde{c}_2}{\zeta}}, \text{ and} \tag{38}$$

$$\mathbb{P}(\|X_\infty^{(n)}\| > t) \leq 2^\alpha\frac{c + \epsilon}{t^\alpha} + \frac{2\tilde{c}_1}{tn^{1/2}}\sqrt{\log\frac{\tilde{c}_2}{\zeta}}. \tag{39}$$

This concludes the proof of Theorem 3. $\qquad\qquad\square$

**Proof of results in Section 3.2**

Let us now consider a setting where $f(\cdot, z)$ is not necessarily a quadratic function. As before, we assume the data comes from an unknown data distribution $\mu_z$ on $\mathbb{R}^{d+1}$. Furthermore, with $(\hat{Z}_k^{(n)})_{k \geq 1}$, we denote the i.i.d. random variables associated with the empirical measure $\mu_z^{(n)} = n^{-1} \sum_{j=1}^n \delta_{\hat{Z}_j^{(n)}}$, as defined in (15). We consider the case $b = 1$ for simplicity.

## C.3 Proof of Theorem 6

*Proof.* First, we utilize the tail index limits from [HM21] in the strongly convex setting from Theorem 2. We use $\alpha$ and $\beta$ in order to construct the proof and provide the same argument as in Section 3.1.

Denote by $E_n$ the event on which $(X_k^{(n)})_{k \geq 0}$ has a stationary distribution $\pi^{(n)}$ such that $\mathbb{P}(E_n) \geq 1 - \epsilon_n$. Given $E_n$, let $(Z_k, \hat{Z}_k^{(n)})_{k \geq 1}$ be the sequences of i.i.d. optimal couplings for $\mu_z$ and $\mu_z^{(n)}$. Therefore, by construction, for any $k \in \mathbb{N}$, $\mathbb{E}[\|Z_k - \hat{Z}_k^{(n)}\|] = \mathbf{W}_1(\mu_z, \mu_z^{(n)})$. Based on this sequence, we consider the processes $(X_k)_{k \geq 0}$, $(X_k^{(n)})_{k \geq 0}$, $(Y_k)_{k \geq 0}$ defined by the recursions:

1. $X_{k+1} = X_k - \eta \nabla f(X_k, Z_{k+1})$, where $X_0 \sim \pi$,

2. $X_{k+1}^{(n)} = X_k^{(n)} - \eta \nabla f(X_k^{(n)}, \hat{Z}_{k+1}^{(n)})$, where $X_0^{(n)} \sim \pi^{(n)}$,

3. $Y_{k+1} = Y_k - \eta \nabla f(Y_k, Z_{k+1})$ where $Y_0 = X_0^{(n)}$,

with the unique stationary distributions of $(X_k)_{k \geq 0}$ and $(X_k^{(n)})_{k \geq 0}$ being denoted by $\pi$ and $\pi^{(n)}$ respectively. Note again that $(X_k)_{k \geq 0}$ corresponds to the online SGD recursion (2), and $(X_k^{(n)})_{k \geq 0}$ to the offline SGD recursion (4), both with $b = 1$. Under our assumptions, by the definition of geometric ergodicity, there exist constants $c_\rho > 0$ and $\rho \in (0, 1)$ such that the following inequality holds:

$$\mathbf{W}_1(\mathscr{L}(Y_k), \pi) \leq c_\rho \mathbf{W}_1(\mathscr{L}(Y_0), \pi) e^{-\rho k}, \text{ for any } k \geq 0 . \tag{40}$$

Analogously to (25), we can conclude:

$$\mathbf{W}_1(\pi, \pi^{(n)}) \leq (1 - c_\rho e^{-\rho k})^{-1} \mathbf{W}_1(\mathscr{L}(Y_k), \mathscr{L}(X_k^{(n)})). \tag{41}$$

The proof then utilizes Taylor's expansion with the remainder part in integral form (see, e.g., [SCW20]) in the following way:

$$\nabla f(Y_k, z) - \nabla f(X_k^{(n)}, z) = \mathbf{H}_{k,z}^{(n)}(Y_k - X_k^{(n)}) \tag{42}$$

with $\mathbf{H}_{k,z}^{(n)} := \int_0^1 \nabla^2 f(Y_k - u(Y_k - X_k^{(n)}), z) \, \mathrm{d}u$. Therefore:

$$Y_{k+1} - X_{k+1}^{(n)} = Y_k - X_k^{(n)} - \eta \left( \nabla f(Y_k, Z_{k+1}) - \nabla f(X_k^{(n)}, \hat{Z}_{k+1}^{(n)}) \right)$$

$$= Y_k - X_k^{(n)} - \eta \left( \nabla f(Y_k, Z_{k+1}) - \nabla f(X_k^{(n)}, Z_{k+1}) + \nabla f(X_k^{(n)}, Z_{k+1}) - \nabla f(X_k^{(n)}, \hat{Z}_{k+1}^{(n)}) \right)$$

$$= Y_k - X_k^{(n)} - \eta \mathbf{H}_{k,Z_{k+1}}^{(n)} \left( Y_k - X_k^{(n)} \right) - \eta \left( \nabla f(X_k^{(n)}, Z_{k+1}) - \nabla f(X_k^{(n)}, \hat{Z}_{k+1}^{(n)}) \right)$$

$$= (\mathrm{I}_d - \eta \mathbf{H}_{k,Z_{k+1}}^{(n)})(Y_k - X_k^{(n)}) - \eta \left( \nabla f(X_k^{(n)}, Z_{k+1}) - \nabla f(X_k^{(n)}, \hat{Z}_{k+1}^{(n)}) \right).$$

Now, taking the norm of both sides and using the triangle inequality together with Assumption 1, we can obtain the following:

$$\|Y_{k+1} - X_{k+1}^{(n)}\| \leq \| \mathrm{I}_d - \eta \mathbf{H}_{k,Z_{k+1}}^{(n)} \| \|Y_k - X_k^{(n)}\| + \eta \| \nabla f(X_k^{(n)}, Z_{k+1}) - \nabla f(X_k^{(n)}, \hat{Z}_{k+1}^{(n)}) \|$$

$$\leq \| \mathrm{I}_d - \eta \mathbf{H}_{k,Z_{k+1}}^{(n)} \| \|Y_k - X_k^{(n)}\| + \eta L(\|X_k^{(n)}\| + 1) \|Z_{k+1} - \hat{Z}_{k+1}^{(n)}\|.$$

Now,

$$\| I_d - \eta \int_0^1 \nabla^2 f(y - u(y - x), z) \, du \| = \| \int_0^1 I_d - \eta \nabla^2 f(y - u(y - x), z) \, du \|$$

$$\leq \int_0^1 \left\| I_d - \eta \nabla^2 f(y - u(y - x), z) \right\| du$$

$$\leq \int_0^1 \sup_{x \in \mathbb{R}^d} \left\| I_d - \eta \nabla^2 f(x, z) \right\| du$$

$$= R(z), \tag{43}$$

where again $R(z) = \sup_{x \in \mathbb{R}^d} \left\| I_d - \eta \nabla^2 f(x, z) \right\|$. This allows us to conclude the following:

$$\|Y_{k+1} - X_{k+1}^{(n)}\| \leq R(Z_{k+1})\|Y_k - X_k^{(n)}\| + \eta L(\|X_k^{(n)}\| + 1)\|Z_{k+1} - \hat{Z}_{k+1}^{(n)}\|.$$

Denoting $B_k^{(n)} := \|X_k^{(n)}\| + 1$, we have:

$$\|Y_{k+1} - X_{k+1}^{(n)}\| \leq R(Z_{k+1})\|Y_k - X_k^{(n)}\| + \eta L B_k^{(n)}\|Z_{k+1} - \hat{Z}_{k+1}^{(n)}\|$$

$$\leq R(Z_{k+1})R(Z_k)\|Y_{k-1} - X_{k-1}^{(n)}\| + \eta L R(Z_{k+1}) B_{k-1}^{(n)}\|Z_k - \hat{Z}_k^{(n)}\| + \eta L B_k^{(n)}\|Z_{k+1} - \hat{Z}_{k+1}^{(n)}\|$$

$$\leq \|Y_0 - X_0^{(n)}\| \prod_{i=1}^{k+1} R(Z_i) + \eta L \sum_{i=0}^{k} B_i^{(n)}\|Z_{i+1} - \hat{Z}_{i+1}^{(n)}\| \prod_{j=i+1}^{k} R(Z_{j+1}),$$

where again, for any sequence $(a_i)_{i \geq 0}$, $\prod_{i=j}^{k} a_i = 1$, for $j > k$. Furthermore, as $\beta > 1$ and $\mathbb{E}[R(Z_1)^\beta] = 1$, we have:

$$\delta_R := \mathbb{E}[R(Z_1)] < 1. \tag{44}$$

Now, we can take the expectation given $E_n$ and use that $R(Z_1), ..., R(Z_{k+1})$, and $B_0^{(n)}, \ldots, B_k^{(n)}$ are i.i.d. Furthermore, using (43), (44), and proceeding as in the proof of Theorem 4, we can obtain:

$$\mathbb{E}[\|Y_{k+1} - X_{k+1}^{(n)}\|] \leq \eta L (\mathbb{E}[\|X_0^{(n)}\|] + 1) \mathbb{E}[\|Z_1 - \hat{Z}_1^{(n)}\|] \frac{1}{1 - \delta_R}.$$

Finally, denoting $\mathbb{E}[\|X_0^{(n)}\|] + 1 = c_0$, we obtain that, with probability larger than $1 - \epsilon_n$:

$$\mathbf{W}_1(\pi, \pi^{(n)}) \leq c_0 L (1 - c_\rho e^{-\rho k})^{-1} \frac{\eta}{1 - \delta_R} \mathbf{W}_1(\mu_z, \mu_z^{(n)}).$$

Taking the limit as $k \to \infty$, we can conclude the proof of Theorem 6. $\qquad \square$

## C.4 Proof of Theorem 5

*Proof.* Due to [HM21] we know that there exist $\alpha, \beta$ such that $\beta < \alpha$ and:

$$\limsup_{t \to \infty} t^{\alpha + \epsilon} \mathbb{P}\left(\|X_\infty\| > t\right) > 0, \quad \text{and} \quad \limsup_{t \to \infty} t^{\beta - \epsilon} \mathbb{P}\left(\|X_\infty\| > t\right) < +\infty. \tag{45}$$

Using Lemma 3, we can obtain:

$$\mathbf{W}_1(\pi, \pi^{(n)}) \geq \int_0^\infty \left| \mathbb{P}(\|X_\infty\| > t) - \mathbb{P}(\|X_\infty^{(n)}\| > t) \right| dt$$

$$\geq \int_{t'}^{t''} \left| \mathbb{P}(\|X_\infty\| > t) - \mathbb{P}(\|X_\infty^{(n)}\| > t) \right| dt,$$

where we have used that $\lim_{t \to \infty} \mathbb{P}(\|X_\infty\| > t) = 0$ and $\lim_{t \to \infty} \mathbb{P}(\|X_\infty^{(n)}\| > t) = 0$, in order to select $t'$ and $t''$ large enough for the last inequality to hold. Now, using Theorem 2, we can obtain:

$$\int_{t'}^{t''} \left| \mathbb{P}(\|X_\infty\| > t) - \mathbb{P}(\|X_\infty^{(n)}\| > t) \right| dt \geq \int_{t'}^{t''} \frac{c_\alpha}{t^{\alpha + \epsilon}} \, dt - \int_{t'}^{t''} \mathbb{P}(\|X_\infty^{(n)}\| > t) \, dt$$

$$\geq (t'' - t') \frac{c_\alpha}{(t'')^{\alpha + \epsilon}} - (t'' - t') \mathbb{P}(\|X_\infty^{(n)}\| > t'), \tag{46}$$

for some constant $c_\alpha$. Therefore, by choosing $t'' = 2t'$, we can obtain:

$$\mathbb{P}(\|X_\infty^{(n)}\| > t') \geq \frac{c_\alpha}{(2t')^{\alpha+\epsilon}} - \frac{1}{t'}\mathbf{W}_1(\pi, \pi^{(n)}).$$

Using similar arguments, we can obtain:

$$\begin{aligned}
\mathbf{W}_1(\pi, \pi^{(n)}) &\geq \int_{t'}^{t''} \mathbb{P}(\|X_\infty^{(n)}\| > t)\, \mathrm{d}t - \int_{t'}^{t''} \frac{c_\beta}{t^{\beta-\epsilon}}\, \mathrm{d}t \\
&\geq (t'' - t')\mathbb{P}(\|X_\infty^{(n)}\| > t'') - (t'' - t')\frac{c_\beta}{(t')^{\beta-\epsilon}}.
\end{aligned}$$

for a constant $c_\beta$. Choosing $t'' = 2t'$ and substituting $\frac{1}{2}\bar{t} = t'$, we obtain:

$$\mathbb{P}(\|X_\infty^{(n)}\| > \bar{t}) \leq \frac{2}{\bar{t}}\mathbf{W}_1(\pi, \pi^{(n)}) + \frac{2^{\beta-\epsilon}c_\beta}{\bar{t}^{\beta-\epsilon}}. \tag{47}$$

Using Theorem 6 and (46)-(47), we can conclude the proof of Theorem 5. $\qquad\square$

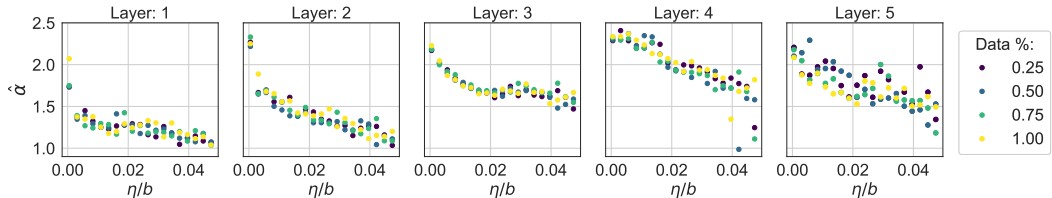

Figure 8: Estimated tail indices, LeNet, CIFAR-10

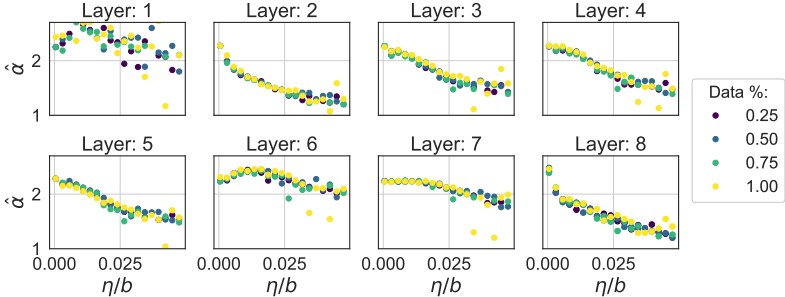

Figure 9: Estimated tail indices, AlexNet, MNIST

## D Further experimental results

In this section, we present the remaining experimental findings.

### D.1 Neural Network Experiments

First, we showcase the remaining plots from Section 4. As outlined in 4.2, we conducted model training for $10,000$ iterations, utilizing the cross-entropy loss and employing the MNIST and CIFAR-10 datasets. The learning rates ranged from $10^{-4}$ to $10^{-1}$, while the batch sizes varied between 1 and 10. We employed offline SGD with a subset of the data amounting to $25\%$, $50\%$, and $75\%$. In Figure 8, we exhibit the estimated tail indices for the LeNet architecture with the CIFAR-10 dataset, while Figure 9 shows the estimated tail indices for the AlexNet architecture implemented with the MNIST dataset.

The inclusion of these plots serves the purpose of completeness, as the overall conclusions remain unchanged; a strong correlation between $\hat{\alpha}^{(n)}$ and $\hat{\alpha}$ is exhibited, as well as a notable correlation between $\hat{\alpha}^{(n)}$'s and the ratio $\eta/b$. These conclusions hold true across all datasets and neural network architectures, providing further substantiation for our theoretical propositions.

### D.2 Further tail examination

From [GSZ21], we have that $\mathbb{P}\left(\|X_\infty\| > t\right) \approx t^{-\alpha}$, so the log-log tail histogram of the stationary distribution should follow a linear line with slope $-\alpha$ in the tails (for large $t$). We first examine the histograms of the estimated stationary distributions for both the linear regression setting (as in Sec. 4.1) and the NN setting (as in Sec. 4.2), analyzing their behavior as the number of samples $n$ increases. These are depicted in Figures 10-11. Then, we re-run the experiments and examine the histograms on a log-log scale to see whether the linear slope $-\alpha$ becomes more apparent. For both cases, as $n$ increases, we observe that this heavy-tailed phenomenon becomes increasingly apparent, and the tails follow a clearer linear trend. The linear behavior in the tails can be observed in Figures 12-13.

In the entirety of our experiments, encompassing diverse learning rates and layers within both the linear regression and NN settings, a consistent observation emerges: an increase in the number of samples employed in offline SGD leads to a convergence of behavior towards that of online SGD. As previously mentioned in Section 1, while we do not anticipate an *exact* power-law tailed behavior, it is noteworthy that the tails exhibit progressively more characteristics resembling a power-law

distribution. Specifically, the log-log plots demonstrate a linear trend in the tails. This empirical observation aligns with our theoretical findings, again reinforcing the consistency between the two.

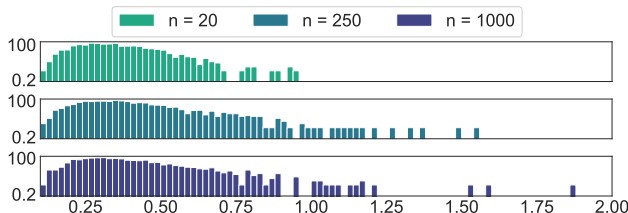

Figure 10: Histograms of the parameter norms for linear regression with Gaussian data

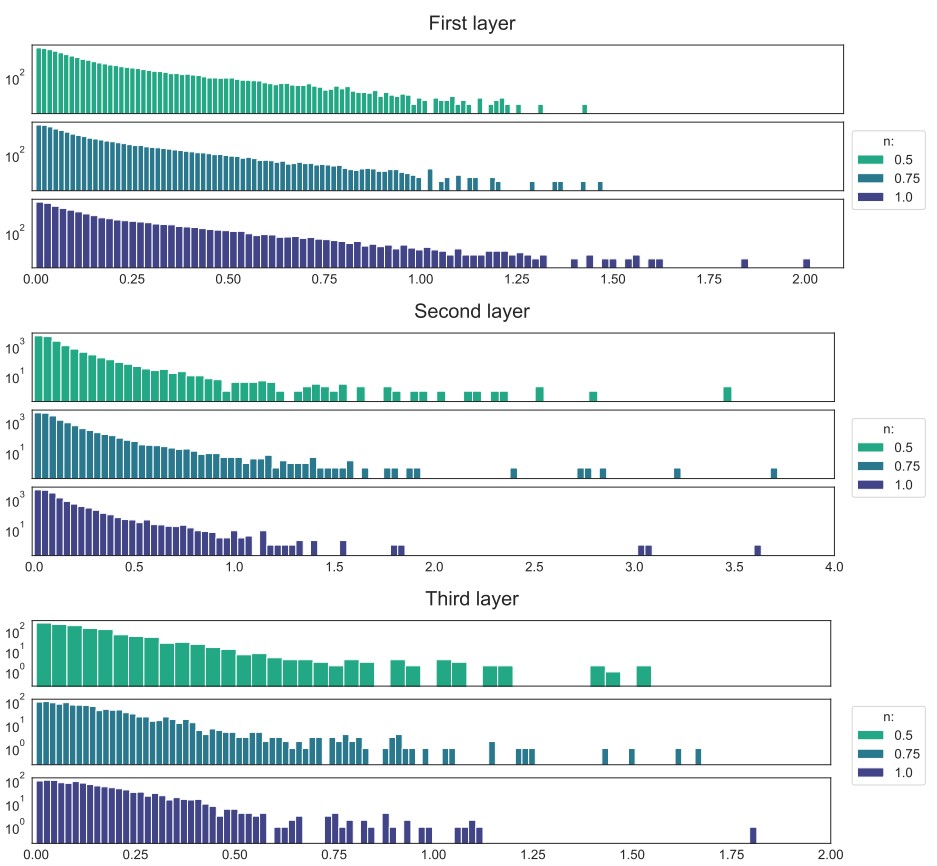

Figure 11: Histograms of the weight norms for FC on MNIST dataset

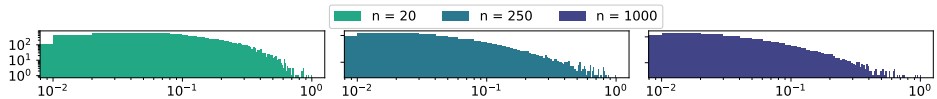

Figure 12: Log-log histograms of the parameter norms for linear regression with Gaussian data

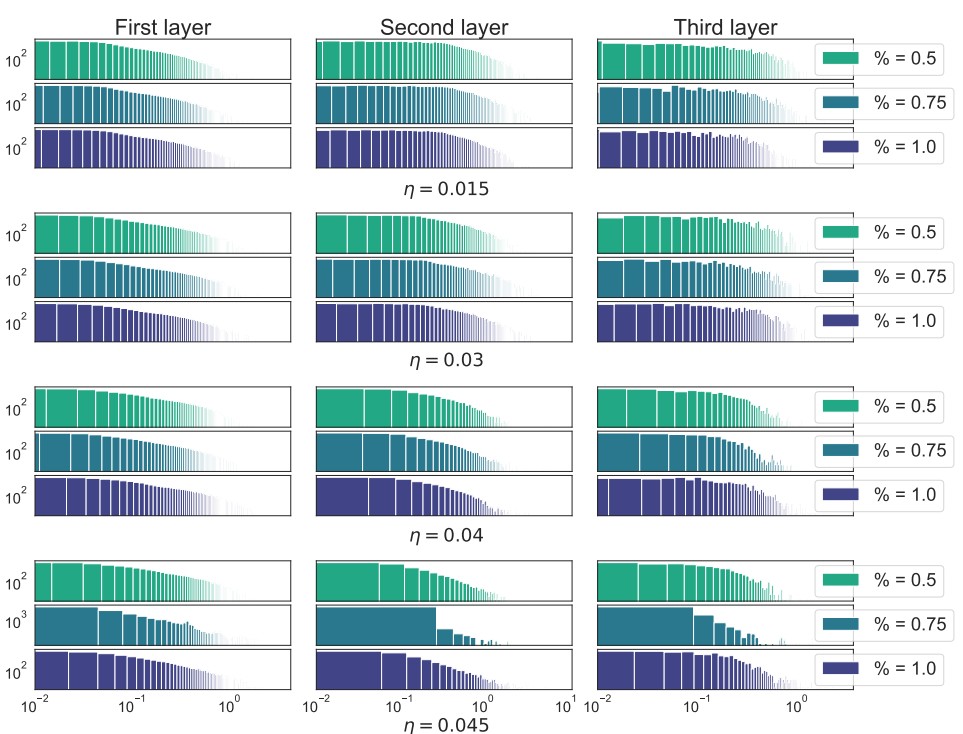

Figure 13: Log-log histograms of the weight norms for FC on MNIST dataset

# E  Estimator choice justification

Analytical computation of the true tail exponent, even in the linear regression setting, is, to our knowledge, unfortunately not possible. Therefore, an estimator choice is required. In this section, we provide our rationale for selecting the tail estimator from [MMO15] by highlighting its strengths.

The main arguments are as follows: *(i)* the estimator's theoretical framework is established through its convergence in distribution to the true tail index (via a Central Limit Theorem result, detailed in Theorem 2.3 [MMO15]), which is further complemented by its demonstrated asymptotic consistency (as per Corollary 2.4 [MMO15]). *(ii)* The estimator has already been used in various articles and its qualities have been thoroughly examined. For example, this can be observed in Figure 14 (taken from [SSG19]), which, in our opinion, contains convincing estimation results (e.g., small error bars regardless of the magnitude of the true tail index).

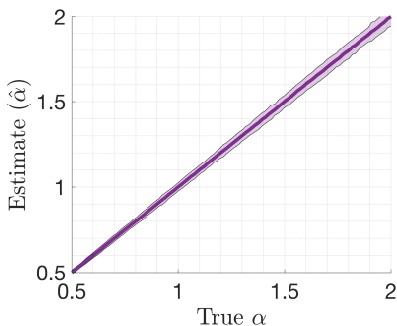

Figure 14: Evaluation of the estimator from [MMO15]. Figure is directly taken from [SSG19].

# F  Strongly Convex Problem Example

Consider the following one-dimensional logistic regression where the regularization parameter $\lambda \sim Exp(\mu)$, i.e., it follows an Exponential distribution with mean $1/\mu$ - its probability density function is as follows:

$$f(\lambda; \mu) = \begin{cases} \mu^{-1} e^{-\lambda/\mu} & \lambda \geq 0 \\ 0 & \lambda < 0 \end{cases}.$$

As before, let $(a_i, q_i)_{i \geq 1}$ be i.i.d. random variables in $\mathbb{R}^2$, such that $z_i \equiv (a_i, q_i)$. In this scenario, where $a \sim N(0, \sigma^2)$ and $y \in \{0, 1\}$, the loss function equals:

$$\ell(x, z) = -y \ln \left( \frac{1}{e^{-ax} + 1} \right) - (1 - y) \ln \left( 1 - \frac{1}{e^{-ax} + 1} \right) + \frac{1}{2} \lambda \|x\|^2.$$

Now, the second derivative of the loss with respect to the parameter $x$ equals $\nabla^2 \ell(x, z) = \frac{a^2 e^{ax}}{(e^{ax}+1)^2} + \lambda$. Note that $|\nabla^2 \ell(x, z)| \leq a^2/4 + \lambda, \forall x \in X$. In the SGD case, from [HM21], we have $r(z) = \liminf_{\|x\| \to \infty} \sigma_{\min} \left( \mathrm{I}_d - \gamma \nabla^2 \ell(x, z) \right)$ and $R(z) = \sup_x \left\| \mathrm{I}_d - \gamma \nabla^2 \ell(x, z) \right\|$. In other words, as we are in the one-dimensional setting, we have:

$$r(z) = \liminf_{|x| \to \infty} \left| 1 - \gamma \lambda - \gamma \frac{a^2 e^{ax}}{(e^{ax} + 1)^2} \right| = |1 - \gamma \lambda|, \text{ and}$$

$$R(z) = \sup_x \left| 1 - \gamma \lambda - \gamma \frac{a^2 e^{ax}}{(e^{ax} + 1)^2} \right| = \max \left( |1 - \gamma \lambda|, |1 - \gamma \lambda - \gamma \frac{a^2}{4}| \right).$$

Now, we require $\mathbb{P}(r(Z_1) > 1) > 0$ and $\mathbb{E}[R(Z_1)] < 1$. As $\lambda \sim Exp(\mu)$, we have that $\mathbb{P}(r(Z_1) > 1) > 0$. For the latter condition, we can calculate the required expectation:

$$\mathbb{E}|1 - \gamma\lambda| = \int_0^{\frac{1}{\gamma}} (1 - \gamma\lambda)\mu e^{-\mu\lambda}\, d\lambda + \int_{\frac{1}{\gamma}}^{\infty} (\gamma\lambda - 1)\mu e^{-\mu\lambda}\, d\lambda$$

$$= -\frac{(\gamma - \mu) - \gamma e^{-\frac{\mu}{\gamma}}}{\mu} + \frac{\gamma e^{-\frac{\mu}{\gamma}}}{\mu}$$

$$= \frac{2\gamma e^{-\frac{\mu}{\gamma}} - \gamma}{\mu} + 1.$$

Furthermore, we have:

$$\mathbb{E}\left|1 - \gamma\frac{x^2}{4} - \gamma\lambda\right| = \mathbb{E}\int_0^{1-\frac{\gamma x^2}{4}} \left(1 - \gamma\frac{x^2}{4} - \gamma\lambda\right)\mu e^{-\mu\lambda}\, d\lambda + \mathbb{E}\int_{1-\frac{\gamma x^2}{4}}^{\infty} \left(\gamma\frac{x^2}{4} + \gamma\lambda - 1\right)\mu e^{-\mu\lambda}\, d\lambda$$

$$= \mathbb{E}\left[\frac{2\gamma}{\mu}e^{\frac{\mu}{\gamma}(1-\frac{\gamma x^2}{4})} + 1 - \frac{\gamma}{\mu} - \frac{\gamma x^2}{4}\right]$$

$$= \int_{-\infty}^{\infty} \left(\frac{2\gamma}{\mu}e^{\frac{\mu}{\gamma}(1-\frac{\gamma x^2}{4})} + 1 - \frac{\gamma}{\mu} - \frac{\gamma x^2}{4}\right)\frac{e^{-\frac{1}{2\sigma^2}x^2}}{\sqrt{2\pi\sigma^2}}\, dx$$

$$= \frac{2\sqrt{2}\gamma}{\mu\sqrt{2 - \sigma^2\mu}}e^{-\frac{\mu}{\gamma}} - \gamma\left(\frac{1}{\mu} + \frac{\sigma^2}{4}\right) + 1$$

We can see that, for example, both expressions are less than 1 in modulus for $\mu = 0.1$, $\sigma^2 = 1$, and $\gamma = 0.1$. It is important to highlight that the aforementioned calculations necessitate the condition $\frac{1}{2\sigma^2} - \frac{\mu}{4} > 0$, or equivalently $\sigma^2\mu < 2$. This condition can be interpreted as a stability criterion, indicating that the mean of the penalization term $\lambda$ must increase as the variance of the data grows. In other words, a larger variance necessitates a higher mean value for $\lambda$.

