# OpenReview forum: "Approximate Heavy Tails in Offline (Multi-Pass) Stochastic Gradient Descent"
_NeurIPS.cc/2023/Conference — NeurIPS 2023 spotlight_

### Official Review · Reviewer_dWGg · 2023-06-26

**Soundness:** 4 excellent
**Presentation:** 4 excellent
**Contribution:** 4 excellent
**Rating:** 7
**Confidence:** 3

**Summary:**

This paper investigates the approximate heavy-tailed behavior of stochastic gradient descent (SGD) in local minima in practical settings and its correlation with overall performance. It shows that the stationary distribution of offline SGD exhibits approximate power-law distribution, and the approximation error is controlled by how fast the empirical distribution of the training data converges to the true underlying data distribution in the Wasserstein metric. The main contribution is to fill the gap in understanding the underlying mechanism generating the reported heavy-tailed behavior in practical settings, where the amount of training data is finite. The paper also proves nonasymptotic Wasserstein convergence bounds for offline SGD to online SGD as the number of data points increases. The theory is verified with experiments

**Strengths:**

- **Originality**: The paper investigates the heavy-tailed behavior of SGD in practical settings where the amount of data is finite, which has not been well understood before. It proves a nonasymptotic Wasserstein convergence bound for offline SGD which reduces to online SGD as the number of data points increases. The difference from a true heavy-tailed distribution with a finite number of data is given by the Wasserstein distance.
- **Quality**: The paper provides rigorous theoretical analysis and proof to support its claims. It also conducts experiments to verify its theory.
- **Clarity**: The paper is well-written and easy to follow, with clear explanations of technical terms and concepts. The authors also provide many clear schematic illustrations to help readers understand their concepts and results.
- **Significance**: The theory of this paper is of both theoretical interest and direct practical significance. The results are easy to understand and bounds on the distance from a true heavy-tailed distribution are given explicitly with a clear form.

**Weaknesses:**

- Possible evaluation method for the values of estimated tail indices is not provided. In Figure 5, the estimated tail indices depend on the step size $\eta$, the batch size $b$, or their ratio $\eta/b$, and of course the number of data $n$. It will be strong if an evaluation method for the tail indices, even approximately, can be provided.

**Questions:**

- In Line 267, the authors state that the tail indices depend on the ratio of the step size $\eta$ to the batch size $b$. However, a mere dependence on the ratio $\eta/b$ has been found broken in online SGD with a large $\eta$ as discussed in Sec.6.6 of Ref.[1]. A modification that is linear in the batch size $b$ is found for a simple 1-dimensional example. Could the authors elaborate on this result?

- Still in Sec.6.6 of Ref.[1], the tail index is estimated with an explicit expression in terms of $\eta$ and $b$ for a 1D example. Is it possible to compare the estimated tail indices predicted by your theorem with the explicit expression given in Ref.[1]?

[1] **Strength of Minibatch Noise in SGD**, Liu Ziyin, Kangqiao Liu, Takashi Mori, Masahito Ueda, *ICLR 2022*

**Limitations:**

The authors adequately addressed the limitations.

---

> ### Author Rebuttal · Authors · 2023-08-09
>
> Thank you for taking the time to invest in our paper and for the encouraging feedback. In the following, we try to address your remaining concerns. However, before proceeding, we would also like to thank you for pointing out the interesting and relevant paper. We apologize for not including it previously (the paper escaped our notice) - we will properly cite it in the next version and add it to the related work section.
>
> > Possible evaluation method for the values of estimated tail indices is not provided. In Figure 5, the estimated tail indices depend on the step size $\eta$, the batch size $b$, or their ratio $\eta/b$, and of course the number of data $n$. It will be strong if an evaluation method for the tail indices, even approximately, can be provided.
>
> As the analytical computation of the true tail exponent is not possible, we were unfortunately not able to perform such an evaluation. Instead, we utilized an estimator from [1], previously employed in [2, 3], which has been demonstrated to be asymptotically consistent (as per Corollary 2.4), ensuring that as the number of samples it uses increases, the estimate will converge to its true value. Furthermore, an empirical evaluation of this estimator has already been undertaken in [3] (we invite the reader to examine their Figure 3(b), which can also be found in the attached PDF in the response to all reviewers), which shows encouraging results. We will mention this in the next version clearly.
>
> > In Line 267, the authors state that the tail indices depend on the ratio of the step size $\eta$ to the batch size $b$. However, a mere dependence on the ratio $\eta/b$ has been found broken in online SGD with a large $\eta$ as discussed in Sec.6.6 of Ref.[1]. A modification that is linear in the batch size $b$ is found for a simple 1-dimensional example. Could the authors elaborate on this result?
>
> While prior studies on heavy-tailed phenomena have explored the tail relationship with the ratio $\eta/b$, you are absolutely correct in pointing out that this dependence may not exhibit a monotonic behavior. In our experiments, we prioritized replicating the NN experimental setup outlined in [2] in order to conduct a comparative analysis between offline SGD with varying $n$ and its online counterpart.
>
> However, the validity of your remark remains: we suspect that the reason why we still observe a monotonic relation between the tail exponent and the eta/b ratio might be that the range that we used for $b$ is too restrictive that the discrepancy result you mentioned does not kick in. We will mention this as a footnote and cite the corresponding part of the paper you mentioned. Nonetheless, our primary focus is to convey the increasing similarity in behavior between offline and online SGD (with the increase in sample size $n$), and we would prefer to avoid the discussion about the monotonicity of the tail exponent with respect to the $\eta/b$ ratio.
>
> > Still in Sec.6.6 of Ref.[1], the tail index is estimated with an explicit expression in terms of $\eta$ and $b$ for a 1D example. Is it possible to compare the estimated tail indices predicted by your theorem with the explicit expression given in Ref.[1]?
>
> Although the expression discovered in [4] is highly captivating, particularly due to its correction term (which is absent in the analysis of [2]), it is important to emphasize that our theoretical findings are strictly orthogonal to estimating the tail index $\alpha$ or the behavior of $\alpha$ with respect to $\eta$, $b$, and their ratio. This relation was established in an earlier paper by Gurbuzbalaban et al [2]. Rather, our analysis focuses on characterizing the dissimilarities in behavior between online and offline SGD. In this respect, our theorems do not predict the tail index directly; they say the tail behavior of offline SGD will become increasingly power-law. Hence, the comparison of their estimate and the article you mentioned would be rather unrelated to our problematic (yet interesting in its correct scope), and we would like to avoid it in order not to convolute our message.
>
> If you have any remaining questions, we look forward to them and remain at your disposal.
>
>
> [1] Mohammad Mohammadi, Adel Mohammadpour, and Hiroaki Ogata. On the tail index and the spectral measure of multivariate α-stable distributions. Metrika,381 78(5):549–561, 2015
>
> [2] Mert Gurbuzbalaban, Umut Simsekli, and Lingjiong Zhu. The heavy-tail phenomenon in SGD, ICML 2021
>
> [3] Umut Simsekli, Levent Sagun, and Mert Gurbuzbalaban. A tail-index analysis of stochastic gradient noise in deep neural networks, ICML 2019
>
> [4] Strength of Minibatch Noise in SGD, Liu Ziyin, Kangqiao Liu, Takashi Mori, Masahito Ueda, ICLR 2022

---

> > ### Comment · Reviewer_dWGg · 2023-08-19
> >
> > Your reply addresses my questions and comments. Thank you very much. I will keep my score as "Accept". I do not raise my score because a possible estimation of the tail index is beyond the scope of this paper. Anyway, I like your idea of using the Wasserstein distance.

---

> > > ### Author Response · Authors · 2023-08-19
> > >
> > > Thank you very much for going over our rebuttal.

---

### Official Review · Reviewer_t9h8 · 2023-06-30

**Soundness:** 3 good
**Presentation:** 4 excellent
**Contribution:** 3 good
**Rating:** 6
**Confidence:** 2

**Summary:**

This manuscript considers the problem of multi-pass stochastic gradient descent with a finite batch-size and strongly convex objective. The key result is to show that when the stationary distribution of the parameters is heavy-tailed in the infinite data (one-pass) limit, then the stationary distribution of the finite batch limit is approximately heavy-tailed, in the sense of having a heavy-tailed component plus a correction depending on the 1-Wasserstein distance between empirical and population data distribution (hence decaying at worst as $\sim n^{-1/2}$). The authors also present numerical experiments that suggest the relevance of this theoretical result to more practical settings.


**Strengths:**

Understanding SGD in its different flavours is an important problem in theoretical machine learning, and results on multi-pass SGD are scarce compared to the one-pass case. Therefore, the contribution is timely and significant. The technical part is sound. Moreover, the paper is well-written and the thread is relatively easy to follow: the motivation and goals are clearly stated and the main result addresses them. Finally, the numerical simulations supporting a broader scope of applicability are nice.

**Weaknesses:**

There are two immediate shortcomings of the work. First, the setting is rather restricted (strongly convex goals). Second, while the authors cite related literature connecting heavy-tails with generalization, this is not explicitly explored in the context of the paper. Overall, one question that remains in the end of the reading is: the stationary distribution of the weights are approximately heavy-tailed, but so what?

**Questions:**

- **[Q1]**: As a reader who is not very familiar with this line of work ([HM21, HSKM22, MM], etc.) I wonder how to conciliate these results to the classical literature [Fabian 1968, Kushner 1981, Pflug 1986] showing asymptotic normality of the stationary distribution under certain conditions. For instance, considering online SGD in the simple least-squares setting. Since the gradient is proportional to the residual $r_{i} = y_{i}-\langle a_{i}, x\rangle$:
$$
  x^{k+1}  = x^{k} + \gamma r_{k} a_{k}
$$

if the residuals are Gaussian close to the minimum (e.g. $y_{i}=\angle a_{i}, x_{\star}\rangle + z_{i}$ for some $x_{\star}$ and $z_{i}\sim\mathcal{N}(0,\sigma^2)$), won't the stationary distribution be Gaussian? What I am missing?

- **[Q2]**: Theorems 2 and 3 have an explicit dependence on the sample size $n$, but how do they depend on the batch size b?

**Minor comments**

- The following sentences are strong and misleading:
> *Previous works on this problem only considered, up to our knowledge, online (also called single-pass) SGD, which requires an infinite amount of data.*

> *However, all these results rely on exact heavy tails, which, to our current knowledge, can only occur in the online SGD regime where there is access to an infinite sequence of data points.*

While it is true that in one-pass SGD the maximum number of steps is limited by the availability of data, this doesn't necessarily means that it requires an infinite amount of data. Note that the number of steps required generically depend on the task under consideration, and in some practical scenarios where data is abundant only a few epochs are required, see e.g. Table 2.2. of [[Kaplan et al. 2020]](https://arxiv.org/pdf/2005.14165.pdf). The authors even acknowledge this fact in L29-30.

- Please add the details of the plots in the caption in Fig. 2 & 5: the reported tail indices correspond to what simulation?

- L128: You mean $[n] = \{1,\cdots, n\}$?

**References**

[[Fabian 1968]](https://projecteuclid.org/journals/annals-of-mathematical-statistics/volume-39/issue-4/On-Asymptotic-Normality-in-Stochastic-Approximation/10.1214/aoms/1177698258.full) Vaclav Fabian. *On Asymptotic Normality in Stochastic Approximation*. Ann. Math. Statist. 39 (4) 1327 - 1332, August, 1968. https://doi.org/10.1214/aoms/1177698258

[[Kushner 1981]](https://epubs.siam.org/doi/10.1137/0319007) Harold J. Kushner and Hai Huang. *Asymptotic Properties of Stochastic Approximations with Constant Coefficients*. SIAM Journal on Control and Optimization, 19(1), 1981

[[Pflug 1986]](https://epubs.siam.org/doi/10.1137/0324039) Georg Ch. Pflug. *Stochastic minimization with constant step-size: Asymptotic laws*. SIAM Journal on Control and Optimization, 24(4):655–666, 1986


**Limitations:**

I have discussed some of the limitations in the "Weaknesses" part.

---

> ### Author Rebuttal · Authors · 2023-08-09
>
> We are grateful for your in-depth review of the paper and the feedback you shared. In the following, we aim to respond to your concerns.
>
> > First, the setting is rather restricted (strongly convex goals)
>
> We agree that considering strongly convex objective functions may seem restricted. However, this setting is in general privileged and considered interesting to study local SGD behaviour, even in the non-convex setting. As a result, analyses of stochastic optimization algorithms in strongly convex settings still attract significant attention, especially in under-examined contexts, such as ours.
>
> That being said, we believe that it is possible to extend our results to non-convex settings, where loss functions satisfy the so-called “dissipativity” property. This condition would essentially let us obtain contractions in the Wasserstein distance and would include several practical settings such as NNs [4] and functions strongly convex outside a bounded region (see [3, Sec. 4] for more examples). However, we leave analyses of such objectives as future work as it would make the paper much more technical and jeopardize the clarity of our main message. We will mention this future direction explicitly in the next version of the paper.
>
> > The question remains: the stationary distribution of the weights are approximately heavy-tailed, but so what?
>
> This is a fair question. As you mentioned, the connection between heavy tails and generalization has been established in certain “sterile” settings; however, an explicit connection for heavy tails in the sense of Thm. 1 and 2 is yet to be developed. Nevertheless, there has been empirical evidence that the tail exponents given in Thms. 1 and 2 may have direct links to generalization, (see [5, Fig. 2]). Hence, we believe that our results will play a key role in understanding generalization error of **offline** SGD, once the link between the tail exponent of **online** SGD and generalization is made explicit – which seems to be an easier task as several quantities can be computed more easily in the online setting.
>
> > [Q1]: … I wonder how to conciliate the results to the classical literature showing asymptotic normality of the stationary distribution...
>
> You are right in your comment: **if the step-size is small enough**, the tail index $\alpha$ will be larger than 2, hence admitting a finite variance. In such cases, one can show that the (properly) averaged iterates will converge to a Gaussian law, which is the main message of the papers you mentioned. However, there is a caveat:
> 1) If the step-size is large enough, $\alpha$ can be strictly smaller than 2, and the averaged iterates will converge to an $\alpha$-stable law, where the studies you mentioned cannot be applied. This has been proven in [1, Corollary 11].
> 2) Even when $\alpha>2$, the iterates will have a power-law distribution, but the behavior can vastly differ to a Gaussian.
>
> Given the surge of interest in large step-sizes and edge-of-stability, we believe that point 1) has significance and illuminates under-explored settings considered in modern practical ML settings. We will mention this point more clearly in the next version.
>
> > [Q2]: Theorems 2 and 3 have an explicit dependence on the sample size n, but how do they depend on the batch size b?
>
> Firstly, due to Thm. 1 the tail index strictly increases in batch size $b$ (see Thm. 4 [1]). Furthermore, Thm. 2 establishes that reducing $b$ leads to heavier tail exponents (see Sec. 4 [2]).
>
> Concerning our presented results, the linear regression bound error terms (see Thm. 3) do not depend on $b$. This arises from the quadratic loss (i.e., linear derivative), linearity of expectation, and Minkowski's inequality (see Appendix C2 (L496) in the supplement).
>
> For Thm. 5, in order to maintain clarity and facilitate a better understanding of the proofs, we have set $b=1$ (as emphasized in L218). We believe increasing $b$ would introduce further notational complexity, potentially compromising the overall presentation.
>
> > While in one-pass SGD the maximum number of steps is limited by data availability, this doesn't necessarily mean it requires an infinite amount of data.
>
> We agree with you: in terms of convergence analysis, an infinite amount of data might not be needed for SGD to find a local minimum. However, to provide a rigorous theoretical analysis of SGD’s tail exponent, existing theory, unfortunately, requires an infinite amount of data.
>
> Nonetheless, your concern is valid: the original sentence might have appeared too assertive. Therefore, we will reword it to highlight that the emergence of heavy tails in theoretical findings is contingent upon access to an infinite amount of data.
>
> > Please add the details of the plots in the caption in Fig. 2 & 5: the reported tail indices correspond to what simulation?
>
> Figures 2 and 5 align with the experiments emphasized on L268. However, it appears certain plots lacked clarity; therefore, we nomenclated the relevant paragraph, referencing it in the corresponding figures. Additionally, we provided further details to the plots. Finally, we provided new sets of plots to Fig. 2 and 3. The new figures and findings can be found in the PDF attached to our general response.
>
> We hope these efforts will enhance the lucidity of our results.
>
> > L128: You mean [n]=1,⋯,n ?
>
> You are right. To make it more visible, we will update it as $[n] = ${$1, \dots, n $}.
>
> [1] Gurbuzbalaban, et al. The heavy-tail phenomenon in SGD, ICML 2021
>
> [2] Hodgkinson and Mahoney. Multiplicative noise and heavy tails in stochastic optimization, ICML 2021
>
> [3] Erdogdu et al. "Convergence of Langevin Monte Carlo in chi-squared and Rényi divergence." AISTATS, 2022
>
> [4] Akiyama, and Suzuki. "Excess Risk of Two-Layer ReLU Neural Networks in Teacher-Student Settings and its Superiority to Kernel Methods.'' arXiv:2205.14818
>
> [5] Raj, Anant, et al. "Algorithmic stability of heavy-tailed stochastic gradient descent on least squares." ALT, 2023

---

> > ### Comment · Reviewer_t9h8 · 2023-08-18
> >
> > Thank you for the detailed answer and for welcoming my suggestions.
> >
> > > *If the step-size is large enough, $\alpha$ can be strictly smaller than 2, and the averaged iterates will converge to an $\alpha$-stable law, where the studies you mentioned cannot be applied. This has been proven in [1, Corollary 11].*
> >
> > How large is large enough in this simple least-squares example? To my knowledge for $\gamma\geq 2$ the mean-squared error $||x_{\star} - x_{k}||^{2}_{2} \to\infty$ as $k\to\infty$, so am confused on how this can correlate with generalization in this case.

---

> > > ### Author Response · Authors · 2023-08-18
> > >
> > > Thank you for going over our rebuttal.
> > >
> > > This is indeed a good question; however, we shall mention that the question rather concerns prior work and it is quite orthogonal to our contributions.
> > >
> > > Nevertheless, let us try to provide an answer, which essentially relies on multiple "surprising" facts. We will try to summarize the picture as follows.
> > >
> > > 1. For the case of linear regression with Gaussian data, the step-size required for the iterates to converge to a distribution with infinite variance is given in arxiv:2006.04740 Proposition 5. For general loss functions, we are not aware such an explicit result.
> > >
> > > 2. You are right, when the step-size is large the iterates may diverge in L2 (as you mentioned); however, they can still converge in Lp with p < alpha < 2 (not to a point though, to a random vector instead), see arxiv:2006.04740 Theorem 8 and Theorem 6  (or you can see arxiv:2102.10346 Theorem 3 for a similar outcome in a different setting).
> > >
> > > 3. As we mentioned in our earlier response, the connection between heavy tails and generalization has been established in certain “sterile” settings. As an example for such a connection, let us consider arxiv:2206.01274 Theorem 4, where the authors showed that even when the original loss value diverges (e.g. your L2 example), the algorithm can generalize well under a "surrogate loss" (for instance the original loss can be L2 but the surrogate loss can be Lp -- this is what they considered in their result). There are other papers, which use bounded surrogate losses -- for instance using 0-1 loss (accuracy) as a surrogate loss whereas the algorithm tries to minimize another loss function (cross entropy etc).
> > >
> > > We hope that this answers your question. We remain at your disposal if there is any further questions.

---

> > > > ### Comment · Reviewer_t9h8 · 2023-08-19
> > > >
> > > > Thank you for the pointers and for the clarification. I will need some time to digest it.
> > > >
> > > > I will keep my score for now and might re-evaluate it after the discussion period with the other reviewers.

---

### Official Review · Reviewer_rBYn · 2023-07-04

**Soundness:** 3 good
**Presentation:** 3 good
**Contribution:** 3 good
**Rating:** 7
**Confidence:** 3

**Summary:**

In this paper, the authors study the heavy-tail distribution for the parameters in offline stochastic gradient descent algorithm (SGD). Theoretical results are provided for a quadratic loss and a strongly convex problem, while numerical results cover more realistic cases such as fully connected NN or CNN. In the theoretical part of the paper, the authors link the tail of the distribution of the offline SGD to the online one. To be more specific, the authors show that the qualitative difference between the online and the offline tail is bounded by the Wasserstein distance between the generating and empirical distribution of the data. The author further bound the Wasserstein distance between the distribution of the online and the offline parameters by the same distance. The theoretical result is qualitatively supported by the numerical experiment.

**Strengths:**

* Originality and significance: the authors extend the analysis of heavy tail in SGD from the online settings to offline settings. Personally, I find it really interesting of having a quantitative non-asymptotic bound of the difference between the two cases. It could make the existing and future online SGD analysis more relevant.
* Quality: In my personal opinion, the authors perform convincing theoretical and numerical analysis.
* Clarity: The paper is well written. The authors present their results in a self-contained way, with multiple examples and intuitions helping illustrating the idea behind their main results. And I appreciate that the code is attached to the paper.

**Weaknesses:**

It appears to me that the fig.1 and fig.3 are not as convincing as the rest of the paper. The axes are unlabeled in these figures. The fact that the distribution has a heavy tail is not very obvious in these figures. For example, to my eye, the n=250 histogram is more power-law comparing to the n=1000 histogram.

The weaknesses and questions are well addressed by the authors in their rebuttal. The authors provided detailed explanations and additional figures.

**Questions:**

I believe that it would be nice if the authors could include results, expressed in numbers, which characterizes the difference between the experimental tail and a real power-law tail.

**Limitations:**

The authors adequately addressed the limitations.

---

> ### Author Rebuttal · Authors · 2023-08-09
>
> We appreciate the encouraging feedback and valuable comments you provided. Moving forward, we will address your primary concern:
>
> > It appears to me that the fig.1 and fig.3 are not as convincing as the rest of the paper. The axes are unlabeled in these figures.
>
> We have taken your feedback into account and therefore carefully labeled all the plots, aiming to improve the clarity and interpretability of the paper.
>
> >  The fact that the distribution has a heavy tail is not very obvious in these figures. For example, to my eye, the n=250 histogram is more power-law comparing to the n=1000 histogram.
>
> In reference to the histograms in Figure 1,  our assertion is that with an increase in the number of data samples $n$ in offline SGD, the distributions exhibit characteristics of heavy-tailed distributions more predominantly. Specifically, we focus on a characteristic property of heavy-tailed distributions: as the tails get heavier, it becomes more likely to observe samples that can be far from the bulk of the distribution (this can be viewed as outliers in a sense).
>
> Accordingly, in Figure 1, we assert that the number of samples located far from the bulk of the distribution increases as the number of available samples $n$ increases.
>
> However, your argument regarding the clarity of the results remains valid. Therefore, we have re-run the experiments and introduced a new set of plots. In Figure 1, we highlighted the mean and mean+2std. of the distribution, as well as marked the samples which exceed the mean+2std threshold. With this, our aim is twofold: firstly, it can be observed that the mean and the std. of the distributions increase as $n$ increases. Secondly, it can be observed that the number of samples that exceed the mean+2std threshold increases, as well as their distance from the threshold. The added figures can be found in the attached PDF file in our response to all reviewers (Fig. 2(a)-(b)).
>
> Regarding Figure 3, we have conducted another set of experiments to provide a clearer analysis of the distributions’ tails. The corresponding plots can also be found in our response to all reviewers (attached PDF, Fig. 1). Specifically, we have run a 1-dimensional linear regression experiment for both offline and online SGD and plotted the corresponding QQ plots of the (estimated) stabilizing distributions. From this, one can observe that:
> * Online SGD with a sufficiently large learning rate exhibits heavy, non-Gaussian tails.
> * Offline SGD exhibits increasingly heavier tails as the sample size $n$ increases.
>
> We hope that these new experiments offer further evidence supporting the notion that the distribution's tail becomes heavier as the number of available samples increases.
>
> >I believe that it would be nice if the authors could include results, expressed in numbers, which characterizes the difference between the experimental tail and a real power-law tail.
>
> Analytical computation of the true tail exponent, even in the linear regression setting, is unfortunately not possible to our knowledge. As we cannot compute the tail exponent, we utilize the estimator from [1], which has been demonstrated to be an asymptotically consistent estimator (as indicated in their Corollary 2.4 in their paper): meaning that as the number of samples we use in the estimator grows, the estimated alpha value will converge to its true value.
>
> As the same estimator has been already used in various articles, some of its qualities have already been demonstrated. We invite you to see [2], Figure 3(b), where the authors empirically demonstrate the accuracy of the estimator on a synthetic task, which we believe is encouraging. In our response to all the reviewers, we have included the aforementioned figure (see Fig. 2(c)), as well as further justification of the estimator choice.
>
> We hope the new experiments and plots have effectively addressed your concerns and we remain at your disposal if there will be further questions.
>
> [1] Mohammad Mohammadi, Adel Mohammadpour, and Hiroaki Ogata. On the tail index and the spectral measure of multivariate α-stable distributions. Metrika,381 78(5):549–561, 2015
>
> [2] Simsekli, Umut, Levent Sagun, and Mert Gurbuzbalaban. "A tail-index analysis of stochastic gradient noise in deep neural networks." International Conference on Machine Learning. PMLR, 2019.

---

> > ### Comment · Reviewer_rBYn · 2023-08-17
> >
> > Thank you for the explanation and additional figures. I have raised my rating to "Accept".

---

> > > ### Author Response · Authors · 2023-08-17
> > >
> > > Thank you very much. We are happy to see that our rebuttal have addressed your concerns.

---

### Official Review · Reviewer_VaqK · 2023-07-07

**Soundness:** 4 excellent
**Presentation:** 4 excellent
**Contribution:** 4 excellent
**Rating:** 8
**Confidence:** 4

**Summary:**

This paper considers offline SGD and proves an approximate power-law tail behavior of the stochastic gradient for strongly convex objectives, confirming the heavy-tail heuristic encountered in practices. Explicit tail estimations are obtained, and as a intermediate result, nonasymptotic Wasserstein convergence results for offline SGD are proved.

**Strengths:**

1. The paper provides a good analysis of the behavior of offline SGD, and contributes to our understanding of heavy-tail phenomena in SGD training.
2. I have gone over the proofs and believe it's basically correct.
3. The paper is well-written and comfortable to read.

**Weaknesses:**

I do not see apparent weaknesses in this paper.

**Questions:**

I have no additional questions.

**Limitations:**

Yes.

---

> ### Author Rebuttal · Authors · 2023-08-09
>
> We are grateful for your encouraging feedback and for taking the time to understand the paper clearly and go through the mathematical proofs. If any questions arise regarding our work, we remain at your disposal.

---

### Author Rebuttal · Authors · 2023-08-09

We want to extend our gratitude to the reviewers for their insightful feedback. In the following, we outline the modifications implemented in response to their input.

**In response to the clarity of the presented figures**:
* Every plot has been carefully labeled and experimental paragraphs nomenclated with the intention of improving the manuscript’s clarity and readability
* We have included new experiments and figures (see Fig. 1 and Fig. 2(a)-(b) in the attached PDF file), emphasizing even further how the tail heaviness of offline SGD iterates increases as the number of samples $n$ increases

More specifically, we have added the following two figures.

Fig. 1 depicts a 1-dimensional linear regression task, connecting how the tail behavior differs between online and offline SGD (with varying $n$). To be precise, the depicted QQ plots of the corresponding (estimated) stabilizing distributions convey the following:
* Online SGD with a sufficiently large learning rate exhibits heavy, non-Gaussian tails.
* Offline SGD exhibits increasingly heavier tails as the sample size $n$ increases

Fig. 2(a)-(b) corresponds to offline SGD applied to a 100-dimensional linear regression task and a NN classification task utilizing a 3-layer fully-connected NN on the MNIST dataset. In both experiments, the following consistent patterns emerge:
* The means and standard deviations of the parameters’ estimated stabilizing distributions in offline SGD increase with $n$
* The quantity and magnitude of the parameters far from the bulk of the distribution increase with $n$

**In response to the validity of the chosen tail estimator**:
* We will add a paragraph in the Appendix that highlights the main strengths of the estimator and justifies our choice for its selection. The main arguments are as follows:
    * The estimator's theoretical framework is established through its convergence in distribution to the true tail index (via a Central Limit Theorem result, detailed in Theorem 2.3 [1]), which is further complemented by its demonstrated asymptotic consistency (as per Corollary 2.4 [1]).
    * The estimator has already been used in various articles and its qualities have been thoroughly examined. This can be observed in Figure 2.(c) in the attached PDF (taken from [2]), which, in our opinion, contains convincing estimation results (e.g., small error bars regardless of the magnitude of the true tail index).

We hope that the aforementioned points convince the reviewers as well as future readers of the estimator choice, a decision that has been endorsed both by our research and preceding academic studies [2,3].

**In response to the remaining points, we**:
* Carefully emphasized that the claims of our paper are predominantly related to offline SGD, in contrast to previous works that consider online SGD
* Included new references and further insights into related work
* Fixed smaller technical notation
* Further elaborated on certain aspects, such as:
    * Extending our results to the non-convex settings with “dissipative” loss functions
    * The significance of our contribution towards linking heavy tails and generalization
    * Discussed potential connections between the true step size and parameters such as $\eta$, $b$, and $n$

[1] Mohammad Mohammadi, Adel Mohammadpour, and Hiroaki Ogata. On the tail index and the spectral measure of multivariate α-stable distributions. Metrika,381 78(5):549–561, 2015

[2] Umut Simsekli, Levent Sagun, and Mert Gurbuzbalaban. A tail-index analysis of stochastic gradient noise in deep neural networks, ICML 2019

[3] Mert Gurbuzbalaban, Umut Simsekli, and Lingjiong Zhu. The heavy-tail phenomenon in SGD, ICML 2021

---

### Decision · Program_Chairs · 2023-09-21

**Decision:**

Accept (spotlight)

**Comment:**

The reviewers all gave the paper strong scores, and it is a new perspective on a fundamental well-studied topic.